# A robust approach for MicroED sample preparation of lipidic cubic phase embedded membrane protein crystals

Michael W. Martynowycz[1,2], Anna Shiriaeva [1,2], Max T. B. Clabbers [1,2], William J. Nicolas[1,2], Sara J. Weaver[1,2], Johan Hattne [1,2] & Tamir Gonen [1,2,3] ✉

Crystallizing G protein-coupled receptors (GPCRs) in lipidic cubic phase (LCP) often yields crystals suited for the cryogenic electron microscopy (cryoEM) method microcrystal electron diffraction (MicroED). However, sample preparation is challenging. Embedded crystals cannot be targeted topologically. Here, we use an integrated fluorescence light microscope (iFLM) inside of a focused ion beam and scanning electron microscope (FIB-SEM) to identify fluorescently labeled GPCR crystals. Crystals are targeted using the iFLM and LCP is milled using a plasma focused ion beam (pFIB). The optimal ion source for preparing biological lamellae is identified using standard crystals of proteinase K. Lamellae prepared using either argon or xenon produced the highest quality data and structures. MicroED data are collected from the milled lamellae and the structures are determined. This study outlines a robust approach to identify and mill membrane protein crystals for MicroED and demonstrates plasma ion-beam milling is a powerful tool for preparing biological lamellae.

G protein-coupled receptors (GPCRs) are membrane proteins critical to physiological functions in the human body[1]. Determining GPCR structures using traditional X-ray crystallography is challenging and typically requires crystallization in lipidic cubic phase (LCP)[2]. Many membrane protein crystals only grow to be a few micrometers in size, and extracting crystals from the viscous LCP is difficult. Structural investigations of GPCRs initially turned to X-ray free electron lasers (XFEL) with injector-based LCP delivery systems[3]. After years of development, this became a tractable approach if large enough crystals are obtained. However, XFEL sources are costly, access is highly competitive, and data processing is difficult. In this approach, many individual crystals are typically used, and data from several thousands are then merged to determine a structure[4]. Single particle cryogenic electron microscopy (cryoEM) is an alternative that does not require crystallization, but the small size of most GPCRs prior to the binding of a signaling partner often makes this approach challenging unless large complexes can be formed[5,6]. Microcrystal electron diffraction

(MicroED) is a cryoEM method for determining three-dimensional structures using nanocrystals, and is ideally suited to determine small membrane protein samples[7]. However, the challenges associated with preparing LCP embedded samples for MicroED experiments have thus far limited the use of this method for these critically important structures.

Recent MicroED investigations have reported structures of membrane proteins in viscous media by focused ion-beam milling and subsequent MicroED data collection. In the case of the functional mutant of the murine voltage dependent ion-channel crystallized in lipid bicelles, optimized blotting and dilution on-grid eventually allowed crystal edges to be identified by FIB-SEM[8]. Zhu et al.[9] demonstrated MicroED data collection from LCP-embedded crystals of proteinase K by converting the LCP to a less viscous mixture using additives that allowed the liquid layer to be easily blotted away. However, this approach failed when tested on crystals of membrane proteins. Polovinkin et al.[10] demonstrated diffraction data from

[1]Howard Hughes Medical Institute, University of California, Los Angeles, CA 90095, USA. [2]Department of Biological Chemistry, University of California, Los Angeles, CA 90095, USA. [3]Department of Physiology, University of California, Los Angeles, CA 90095, USA. ✉e-mail: tgonen@g.ucla.edu

bacteriorhodopsin grown in LCP. In this example, a single bacteriorhodopsin crystal over 50 μm wide was looped, placed on an EM grid, milled using a gallium ion-beam, and electron diffraction confirmed the unit cell. However, no structure was determined in this investigation for various reasons that were not disclosed. We previously determined the structure of the human adenosine receptor $A_{2A}AR$ from a single microcrystal[11]. To make this sample amenable to MicroED data collection, the crystals were grown in syringes to avoid the rapid dehydration observed from looping crystals from a glass plate and transferring them onto an EM grid. Instead, LCP was converted to the sponge phase inside the syringe. This approach allowed the microcrystal mixture to flow more easily and excess material could be blotted away. Grids were made from this sponge phase mixture and blotted using standard protocols. The microcrystals in the blotted sponge phase grids were visible by FIB-SEM and could be thinned by the gallium beam and subsequently structurally characterized by MicroED[12]. In this work, the looped crystals could be kept hydrated using a humidifier, but resulted in thick layers of ice on the grids, and no crystals could be identified in the FIB-SEM. Thus far, this approach has not been successful for other membrane proteins. It is likely that the conversion to the sponge phase may damage the crystals.

Although these advances have made membrane proteins such as GPCRs more accessible to MicroED, two fundamental issues continue to prevent more widespread adoption: locating crystals in thick media and making the sample thin enough for MicroED experiments. All reports of membrane protein structures from milled crystal lamellae have relied on visually identifying the crystals from topological images in the FIB-SEM prior to milling. Moreover, conversion of the LCP to the sponge phase may damage the crystals limiting the usefulness of such an approach. To tackle more challenging structures, methods must be developed to successfully mill unconverted LCP and locate crystals inside of the deposited viscous LCP. A fundamental issue with attempting ion-beam milling of LCP embedded crystals is that this material is exceptionally difficult to mill using a gallium beam. In our hands, the LCP will begin to indent, appear to burn and then deform rather than being removed from the sample (Supplementary Fig. 1, orange arrows). Milling under these conditions is essentially impossible and has prevented milling into thicker LCP areas on the grids. Finding a method to mill away LCP without changing the phase requires a new approach to milling thick samples that does not involve a standard gallium ion beam.

Thinning vitrified biological specimens using a focused ion-beam of gallium ions has become a standard method to prepare samples for electron cryo-tomography (cryoET) and macromolecular MicroED experiments[13–18]. Unfortunately, milling biological specimens using gallium ions has several drawbacks. For example, gallium sources have limited angular intensity and spherical aberrations that limits their use to relatively low ion-beam currents[19]. Lower currents increase the amount of time needed to prepare a sample. Furthermore, the gallium ions used for thinning can compromise the experiment by implanting within the sample during milling[20], and high-energy gallium ions damage the exposed surfaces of the lamellae[21,22]. These damaged surfaces lower the achievable signal-to-noise ratio. Thinning biological specimens using a gallium beam is particularly challenging for embedded samples, where only a handful of usable crystal or cellular lamellae are prepared over an entire day[23] and is slow and inefficient even with automation[24,25].

Plasma focused ion beams (pFIBs) are often used in materials science and room-temperature slice-and-view imaging of plastic embedded samples[26,27]. Plasma sources are preferable to liquid metal ion-sources for rapid sample preparation, because they maintain coherence at higher beam currents[28]. Additionally, some plasma ion sources, such as xenon, have a higher sputter rate than gallium. This suggests that xenon has the potential to mill faster and cause less damage to the sample than gallium. A recent report using hard

materials compared the implantation of ions from various plasma beams after milling tungsten filaments and demonstrated that xenon resulted in the lowest implantation depth and shortest milling times and that oxygen and nitrogen beams lead to oxide and nitride formation within these samples. These reports are in agreement with the stopping range of ions in matter (SRIM); simulations show that higher-Z sources tend to sputter material faster and damage the surfaces less (Supplementary Fig. 2)[21]. This approach has not been tested on biological macromolecules but preparing biological lamellae using a pFIB should potentially be faster, allow milling thicker cells, and increase the signal-to-noise ratio of the subsequently collected data on a TEM. This faster milling along with lower damage might enable creating lamellae of GPCR crystals buried deep within thick, viscous piles of LCP for subsequent MicroED experiments.

Here, we develop methods to create lamellae of vitrified biological material using a pFIB and correlate images in the pFIB-SEM with an integrated fluorescent and light microscope (iFLM). First, we characterize the four available plasma ion sources – xenon, argon, nitrogen, or oxygen – to prepare lamellae of vitrified biological samples at cryogenic temperatures. To quantitatively assess the outcomes, we vitrified microcrystals of proteinase K on EM grids. Microcrystals are machined for each ion source using the same protocols to a target thickness of 300 nm. This roughly corresponds to the inelastic mean free path of electrons accelerated through 300 kV and typically leads to the highest quality data[29]. Grids with milled lamellae were transferred into a cryogenically cooled transmission electron microscope (TEM), and continuous rotation MicroED[30] data were collected in electron counting mode on a direct electron detector[31]. The data quality, images, quantitative and qualitative features between the various ion-beam sources are compared between individual lamellae. Structures are determined from lamellae created using each gas source to compare the quality of the resulting models. Next, we prepare frozen grids containing fluorescently labeled human adenosine receptor containing a BRIL fusion protein in the third intracellular loop and a C-terminal truncation of residues 317 to 412 ($A_{2A}AR$-BRIL-ΔC, hereafter $A_{2A}AR$) in LCP. While crystals are not visible under the thick LCP layer by SEM, they are clear in fluorescence imaging, allowing efficient targeting and milling. Crystals are identified deep within thick piles of LCP using fluorescent microscopy and correlated to images taken by the SEM to precisely target the crystals. Fluorescent images are taken periodically during lamellae preparation. The structure of $A_{2A}AR$ is then determined by MicroED to a resolution of 2.0 Å.

## Results

### A plasma focused ion-beam for vitrified specimens and targeting using fluorescence

A Helios Hydra 5 CX dual-beam (Thermo Fisher) instrument equipped with a cryogenically cooled stage was employed for these investigations. This instrument enables the selection of either xenon, argon, oxygen, or nitrogen ion sources to form a pFIB and an improved SEM column compared to instruments used in prior investigations ("Methods" section)[15,32,33]. The sample stage operates at a 4 mm working distance that roughly corresponds to the coincidence point between the electron and ion beams that are oriented 52° apart. The sample shuttle for this system holds two clipped TEM grids at a pre-tilt of 27°. The system has an integrated fluorescence light microscope (iFLM) with a ×20 objective that provides an imaging field of view of ~370 μm at a working distance of ~600 μm. The light microscope operates in either reflective or fluorescence mode using one of four selectable excitation wavelengths ($\lambda$ = 385, 470, 565, 625 nm). Light microscopy is conducted by a shuttle translation and a 180° rotation within the chamber from the standard imaging and milling orientation. Integration of the light microscope allows for identification of targets and correlative light and electron microscopy (CLEM).

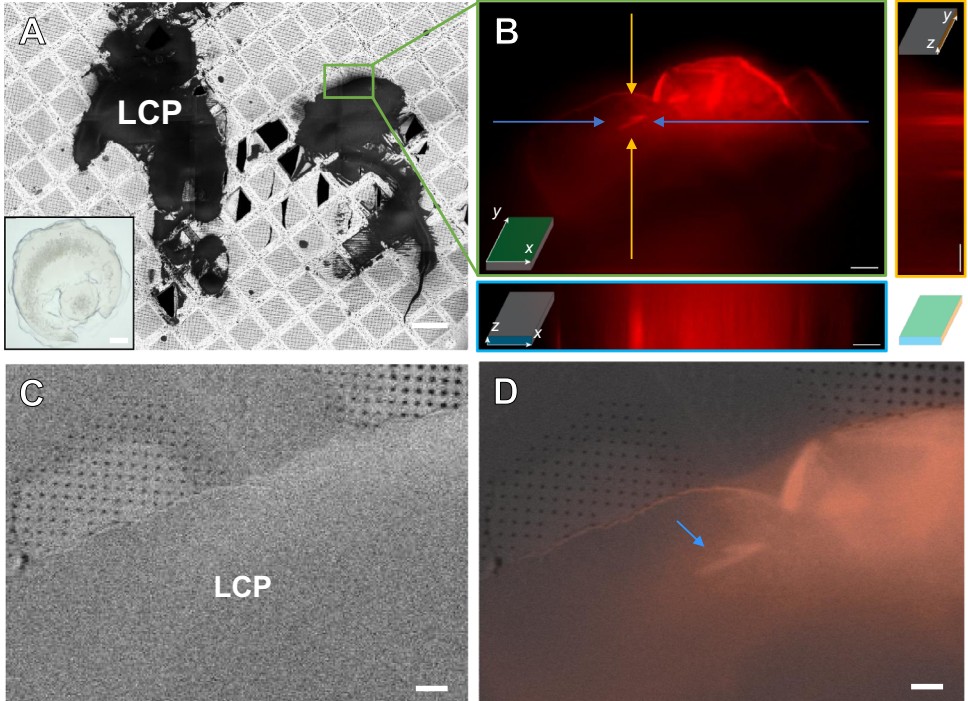

**Fig. 1 | Identification and targeting of fluorescently labeled GPCR microcrystals in thick LCP. A** 500 V SEM image montage of a prepared grid of the human $A_{2A}$ adenosine receptor ($A_{2A}$AR) in lipidic cubic phase (LCP). Inset shows a typical $A_{2A}$AR drop. Scale bar 100 μm. **B** Fluorescent image from a Z-stack taken from the location highlighted from **A** in the green box. The bottom and right panels depict the projection of the stack in either the X-Z or Y-Z planes, respectively. Arrows to the crystal are color coded to their locations in the corresponding projections. Scale bars 25 μm. **C** SEM image of the area in **B** taken after platinum coating. Scale bar 10 μm. **D** Correlative overlay of **B** onto **C** showing the location of the crystals deep in the LCP. Scale bar 10 μm.

The integrated light microscope is designed based on the photon ion electron microscope (PIE scope) as described[34]. Tagging the protein with a fluorophore prior to crystallization enables the unambiguous identification of protein crystals embedded in thick material. In this way, proteins that are buried in thick media can be identified.

We hypothesized that the iFLM could be used to target fluorescently labeled GPCR crystals that were buried in LCP on an EM grid. Protein of $A_{2A}$AR was fluorescently labeled prior to crystallization. The crystals grown in LCP were then spread on a pre-clipped EM grid using a crystallography loop and vitrified in liquid nitrogen. Screening these grids, no crystals were visible topologically using either the SEM or pFIB beams (Fig. 1A). However, translating the stage to the iFLM allowed immediate identification of crystals buried under the surface of the LCP piles (Fig. 1B). Upon identification, a Z-stack of fluorescent images was acquired to target the crystal location (where it is focused best and the fluorescent signal is highest). Overlaying the fluorescence data onto the SEM images allowed pinpointing the crystal coordinates in the pFI-/SEM (Fig. 1C, D). However, several obstacles prevented accurately milling deep into the LCP: the SEM and iFLM data needed to be correlated to the grazing incidence milling pFIB beam, the samples need to be protected from the powerful plasma ion-beam, and the best plasma ion-beam for obtaining the highest quality data had to be determined.

Because the crystals were buried deeply in the thick LCP, we had to determine a way to target them as accurately as possible in the Z-dimension. X- and Y- dimensions are relatively accurate but the Z-dimension (depth) resolution is relatively poor in brightfield cryogenic fluorescent light microscopy. For this we first calibrated the iFLM using fluorescent beads (4 μm Tetraspecs) embedded in a thick matrix of 50% glycerol to mimic the viscosity of the LCP. We alternated between milling and imaging to correlate the iFLM measured depth of the beads and the disappearance depth of the beads measured by the angled view of the pFIB. (Supplementary Fig. 3, and "Methods"

section). Using this method, we were able to reliably target regions of interest buried deep in thick media and make sure that the two means of measurement agree with each other.

Even at low flux, the ion-beam can damage the sample during imaging and milling[18]. Milling is typically conducted at much higher beam currents than imaging[12,14,23]. Although milling is contained to a defined region, the beam is usually much larger than the defined area milled. The spilled over exposures build up at the sample face over the course of the experiment. Additionally, making lamellae with an even thickness requires the front of the sample to be nearly homogenous and smooth. For this purpose, a layer of platinum was deposited to protect the samples using the gas injection system (GIS) at a grazing incidence modified for this use case (Supplementary Fig. 4, Methods). With the milling depth estimation and GIS protection strategy sorted, it was necessary to fully characterize the pFIB sources for milling vitrified biological material.

## Benchmarking different plasma sources

Microcrystals of a serine protease, proteinase K, were grown in batch and vitrified onto TEM grids. The grids were placed into autogrid clips and loaded into the pFIB-SEM and coated with GIS platinum. Crystals were identified on a single grid using SEM imaging (Fig. 2A). Five crystals were milled using each ion source, 20 crystals in total (Supplementary Figs. 5–8). Each crystal was milled at -15°, corresponding to a stage tilt of 4° with an 11° sample pre-tilt and SEM imaging angle of ~67° (Fig. 2A, B). The milling was conducted using pre-defined cleaning cross sections (Methods, Supplementary Table 1). Each lamella was prepared in four steps to a final target thickness of 300 nm (Fig. 2B and Supplementary Table 1). This thickness is roughly the inelastic mean free path of an electron accelerated through a potential of 300 kV, and was previously determined to maximize MicroED data quality[29]. Each set of five lamellae was milled sequentially. The source gas was then switched, the plasma ion-beam aligned, and the next lamellae were

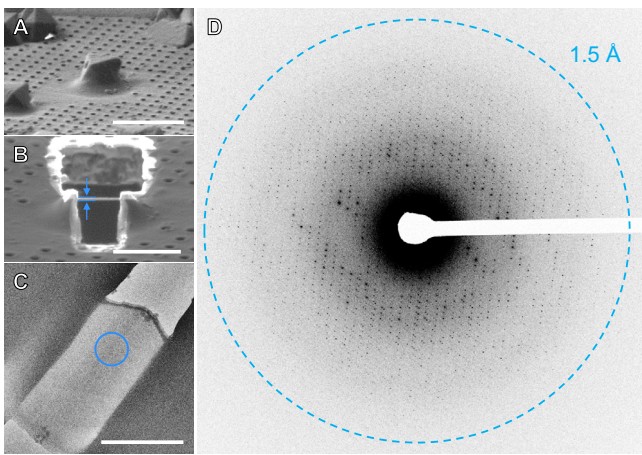

**Fig. 2 | Preparing plasma beam milled lamellae of a protein microcrystal.** Images of a selected serine proteinase microcrystal before (**A**) and after (**B**) thinning the crystal into a thin lamella using a focused ion-beam of argon ions. This lamella showing clear delineation of the platinum layer, crystal, and vitrified media at 2200× in the TEM (**C**). **D** MicroED data corresponding to 4° of data summed together from a direct electron detector. All scale bars 10 μm.

milled. All twenty lamellae across all four gas sources were prepared in a single 10-h shift.

We found differences between the gas sources and categorized the qualities of each gas by the following criteria: milling speed, imaging quality, and success rate (Fig. 3, Supplementary Table 1, and Supplementary Figs. 5–8). By inspection using the ion and electron beams, we found that several crystal lamellae had some signs of cracking, splitting, or being otherwise destroyed during the milling process, notably nitrogen (5/5) and oxygen (4/5) displayed the most damage to the crystals (Supplementary Figs. 5–8). Overall, the milling speeds correlated well with the tabulated and simulated stopping range of ions in matter (Supplementary Fig. 2 and "Methods" section). However, we found that the argon beam could mill at a higher current while preserving the integrity of the lamellae, and could thus mill faster than the xenon beam (Fig. 3A).

Imaging specimens with the plasma ion-beams is similar to using a gallium ion-beam instrument. However, the depth of field was different for each ion source. Adjusting the ion-beam image for any of the plasma sources was more challenging than for gallium sources. The contrast of the images roughly correlates to the mass of the ion—xenon had the best contrast, whereas nitrogen had the worst (Supplementary Figs. 5–8). However, the faster sputter rates for xenon and argon typically made tasks such as focusing the image more challenging, because the area used to focus would rapidly deteriorate at higher beam currents. The oxygen and nitrogen sources have additional blurring due to how the magnetic lenses affect these lighter elements, resulting in 'double images' in both the left-right and up-down directions. The left-right double image can be corrected via direct alignments inside the column. However, the top-down double image could not, and was instead corrected by sticking rare earth magnets to the plasma beam column until sharp images could be obtained (Methods). Lamellae were transferred into a cryogenically cooled TEM for further investigation (Fig. 2C, D).

## MicroED data collection

After cryo-transfer into the TEM, we assessed each lamella by visual inspection of low-dose images taken on a direct electron detector (Fig. 2C and Supplementary Figs. 9–12). Ice contaminations and breakage not observed in the SEM prior to loading in the TEM are attributed to the cryotransfer step. All 20 lamellae sites were identified in the TEM using low magnification imaging. At higher magnifications,

breaks on the far side of 2/5 (1 minor, 1 large) argon milled lamellae became visible along the edges (Supplementary Fig. 10, lamellae #4 and #5). Visual inspection of the unbroken or cracked portions of the milled lamellae was used to assess the degree of curtaining on the surface of each crystal using TEM imaging. In this assessment, all xenon lamellae had evidence of strong curtaining and streaks (Supplementary Fig. 9), most oxygen lamellae had visible curtaining that was less severe than xenon (Supplementary Fig. 12), and argon had the least visible curtaining that we could assess (Fig. 4 and Supplementary Fig. 10). The lamellae milled with nitrogen all contained serious visible pathologies, including a hole through the top of the lamella (Supplementary Fig. 11). Examples of the identified pathologies seen in the TEM are presented in Fig. 4.

Continuous rotation MicroED datasets were collected identically from each lamella in electron counting mode on a Falcon 4 direct electron detector (Fig. 3 and Supplementary Figs. 9–12)[31]. Data were collected from each lamella using the same rotation rate over an identical real-space wedge ("Methods" section). Data were isolated from a 2 μm diameter area using a selected area aperture. In this way, we were able to collect data from nearly all the lamellae (18/20). Maximum intensity projections were calculated to visually inspect the resolution of each dataset prior to processing, since single frames in counting mode contain very little visible signal (Supplementary Figs. 9–12). Electron counting movies were converted to crystallographic format, and then indexed and integrated identically ("Methods" section).

## Evaluating data quality from different plasma sources

Crystallographic intensity statistics were determined after applying a high-resolution cutoff for each dataset where the mean half-set correlation coefficient ($CC_{1/2}$) fell to ~30% (Fig. 3F and Supplementary Figs. 13–16)[35]. Data were merged from lamellae milled from each ion source (Fig. 3C–F) and separately on a per-lamella basis (Supplementary Figs. 13–16) to separate averaged results from individual trends. In terms of crystallographic statistics, we found that the highest average ($<I/\sigma (I)>$) came from lamellae prepared using the argon beam, followed by xenon, oxygen, and nitrogen (Fig. 3C). Completeness was relatively high for each crystal (Fig. 3D). We attribute differences in completeness to variations in crystal orientation on the grid. The mean half-set correlation coefficient ($CC_{1/2}$) and the redundancy corrected merging R factor, $R_{pim}$, showed the same overall trends as ($<I/\sigma (I)>$), where the best results seemingly came from argon, followed by xenon, oxygen, and then nitrogen (Fig. 3E, F). The statistics from oxygen most closely resemble the best results using gallium ions, whereas both argon and xenon data appear to consistently yield better data.

## Structures from plasma milled lamellae

Structures of proteinase K were successfully determined from the merged data of each gas source by molecular replacement (Fig. 5, Table 1, and "Methods" section)[36]. Each structure was refined using the same settings, with calcium and nitrate ions being added manually when found between refinement cycles[37,38]. The resolutions of the lamellae were 1.40, 1.45, 1.50, and 1.80 Å for argon, xenon, oxygen, and nitrogen, respectively (Fig. 3B). This is compared to our prior best result of 1.5 Å using a gallium ion-beam. After the final rounds of refinement, the R-work and R-free for the same experiments were found to be: 13.74/17.35, 13.87/17.70, 16.79/21.21, 16.34/21.38. Surprisingly, the R factors for both argon and xenon milled lamellae were both significantly better than any prior investigation of this protein by MicroED, whereas the R factors for both nitrogen and oxygen were overall similar to those in prior investigations at similar resolutions. The prior best gallium milled structure resulted in an R-work and R-free of 14.95 and 20.46, respectively[31]. The structures determined from plasma milled lamellae all showed well defined side chains and essentially undamaged disulfide bonds (Fig. 5)[39]. As expected, the

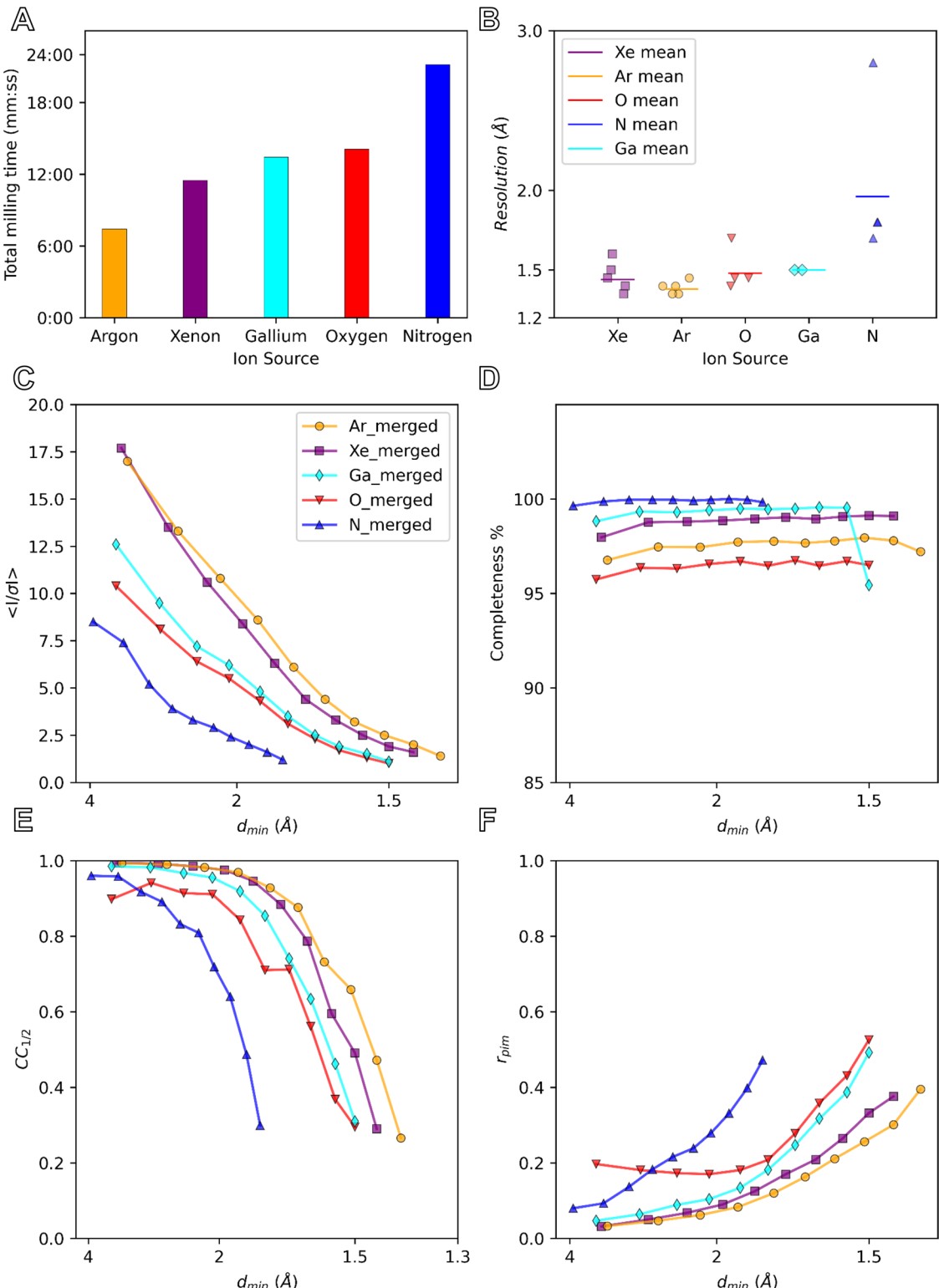

**Fig. 3 | Crystallographic statistics for plasma ion-beam milled lamellae from different sources.** Plots depict the total milling time (**A**), MicroED resolution (**B**), mean signal to noise ratio (<I / σ (I) > ) (**C**), completeness (%) (**D**), mean half-set correlation coefficient (CC$_{1/2}$) (**E**), and merged multiplicity corrected R factor (R$_{pim}$) (**F**) as functions of the d$_{min}$ resolution bins (Å). The merged datasets are solid lines with symbols with xenon (Xe) in purple, argon (Ar) in orange, oxygen (O) in red, nitrogen (N) in blue, and gallium in teal. Solid lines in **B** depict the mean for each gas as indicated, where *n* = 5 for xenon and argon, *n* = 4 for oxygen and nitrogen, and *n* = 2 for gallium milled lamellae.

higher resolution structures of xenon and argon show more resolved waters than the lower resolution model derived from nitrogen milled lamellae. Merging across different ion sources was also explored, and the increased multiplicity resulted in even better structural model to

compare against the individual merged sets ("Best Merge", Table 1, Supplementary Fig. 17, and "Methods" section)[40]. The model derived from oxygen milled lamellae, however, showed a significantly larger number of water molecules than expected based on resolution and

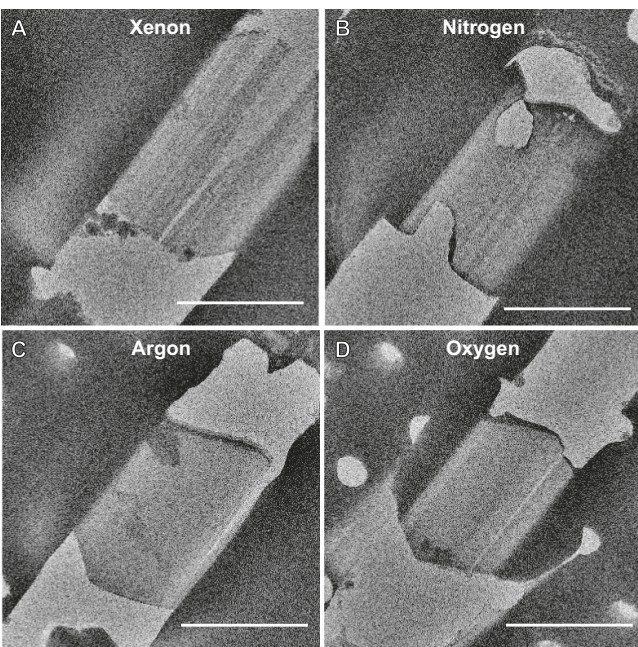

**Fig. 4 | Observed pathologies in proteinase K lamellae prepared by different ion sources on the pFIB. A** xenon lamella showing breaks and deep curtaining (**B**) nitrogen lamella showing perpendicular holes and curtaining (**C**) argon lamella showing a break and topical ice contamination on the leading edge (**D**) oxygen lamella with a break, curtaining, and ice contaminations on the back edge. All scale bars 10 μm.

overall poorer crystallographic refinement, with statistics like the lower-resolution model from the nitrogen beam.

## Targeting membrane protein crystals in LCP by correlated light and electron microscopy

Grids containing $A_{2A}AR$ crystals labeled with one of two different dyes were prepared by looping large amounts of material from crystallization drops in a glass-sandwich plates (Methods). To prevent the rapid degradation of these crystals, the looping was done at high humidity. A 100 μm nylon crystallography loop was used to scoop up a large amount of both LCP and $A_{2A}AR$ microcrystal, gently scraped along the surface of a pre-clipped EM grid, and immediately plunged into liquid nitrogen (Methods). These grids were loaded into the pFIB/SEM at cryogenic conditions. An all-grid atlas was taken of the grid using the SEM at an accelerating voltage of 500 V to better target future positions and increase contrast (Fig. 1A). The grid was then coated in a protective layer of platinum using the GIS similarly to the grids containing proteinase K. From the overview, areas of LCP were identified that were between 5 and 100 μm above the holey carbon film. Crystals are not visible from any angle using either the SEM or pFIB (Fig. 1C). Instead, the stage was translated and inspected using the iFLM using either the reflective mode, or by one of the four wavelengths. Crystals could not be identified using the reflective mode imaging, but the latter was useful to evaluate the topology of the sample and estimate the height of the surface. However, the red and green fluorescent channels successfully identified crystals buried deep within the LCP (Fig. 1B). By taking multiple images over a range of focal distances, we were able to identify the depth of the crystal relative to the surface of the LCP and from the position of the underlying grid bars below. In order to facilitate correlation of the positions in 3D, additional fiducials were created by milling holes into the LCP using the pFIB (Supplementary Fig. 18). The fluorescent and reflective stacks were simultaneously correlated to the X-Y plane of the SEM images (Fig. 1D). The iFLM, SEM, and FIB images were correlated using 3DCT in

order to accurately identify the position of the crystals in 3D, typically within a 1–3 pixels error, or on the order of 100–500 nanometers (Supplementary Fig. 19 and "Methods" section). In this way, the position of an $A_{2A}AR$ crystal was determined in three dimensions to enable targeting of essentially invisible crystals buried in the thick LCP (Methods). The first crystal selected for milling was ~20 μm above the holey carbon film and ~10 μm from the top of the LCP. Additional crystals were nearby in this same pile, but were all directly over a grid bar, rendering them unusable (Fig. 1B and D). On another sample, the crystals were similarly buried between 10 and 50 μm beneath the surface (Fig. 6C, D and Supplementary Fig. 19). Additional crystals were milled on a second grid using a different fluorescent dye (Supplementary Figs. 18 and 19 and "Methods" section).

Lamellae were created from selected $A_{2A}AR$ crystals using either the argon or xenon beams. These were chosen based on the data obtained for proteinase K microcrystals (Fig. 5). Since these crystals were buried deep in piles of LCP, we could use much higher currents for digging the initial trenches compared to the proteinase K crystals (Supplementary Table 2). These were limited by the breaking of the crystals rather than the sputter rate of the ions. Due to the immense size of the LCP occluding the crystal, initial milling was conducted between 4 and 120 nA, currents that would not be feasible for milling frozen samples in a gallium ion system. The current was stepped down as the lamella approached the physical crystal location in Z ("Methods" section). Between each thinning step, one or more fluorescent images were taken at the crystal focal plane to ensure the crystal was not destroyed or over-milled. The final lamella was ~10 μm wide, 250–300 nm thick, and required the removal of at least $10 \times 40 \times 50$ μm of LCP, carbon, and ice from either above or below the suspended crystals ("Methods" section and Supplementary Fig. 19). The plasma ion beam showed no deformation or decoloring of the LCP (Fig. 6A). Due to the increased current and sputter rate, the total milling time on each lamellae being under 1 h, with the majority of time for the experiment being taken by imaging, correlating, deconvolution of iFLM stacks, and checking the sample between milling steps ("Methods" section). A final stack of fluorescent images was taken from the thin lamellae and correlated to an SEM image and confirmed the crystal survived the milling process (Fig. 6B and Supplementary Fig. 19). From the fluorescent image taken at the focal plane of the lamella, we could see the milled crystal appeared sharper in the lamella than the unmilled portion outside of the lamella, indicative of the higher noise from the LCP that was not removed from this area (Fig. 6B, blue versus white arrow). In some cases, the crystalline patch could also be directly identified using the SEM beam after milling (Fig. 6D and Supplementary Fig. 19).

## Screening of the GPCR $A_{2A}AR$ lamellae

Grids containing $A_{2A}AR$ GPCR lamellae were transferred into a cryogenically cooled TEM. The lamellae were located using low magnification imaging and brought to the eucentric position. A single sweep of continuous rotation MicroED data were collected from a real space wedge between either −40° and +40° or −30° and +30° (Fig. 6E, F, "Methods" section). The space group was determined to be $C\,2\,2\,2_1$ with a unit cell of $(a, b, c)$ (Å) = (39.08, 177.51, 139.29) and $(\alpha, \beta, \gamma)$ (°) = (90, 90, 90) (Table 2). The structure was determined by molecular replacement and subsequently refined using electron scattering factors ("Methods" section). We observed strong difference density in the binding pocket corresponding to the bound ligand, ZMA, along with multiple cholesterol and lipid molecules. Many of these were modeled as partial or whole lipids using PDB 4EIY as a reference, and the final densities resolved many lipids and waters (Fig. 7). The overall architecture of the protein was as expected with seven transmembrane helices with a BRIL fusion region in the intracellular region (Fig. 7). We did not observe a sodium binding site in the deep pore of our structure[41]. This structure extended to a resolution of 2.0 Å. This is

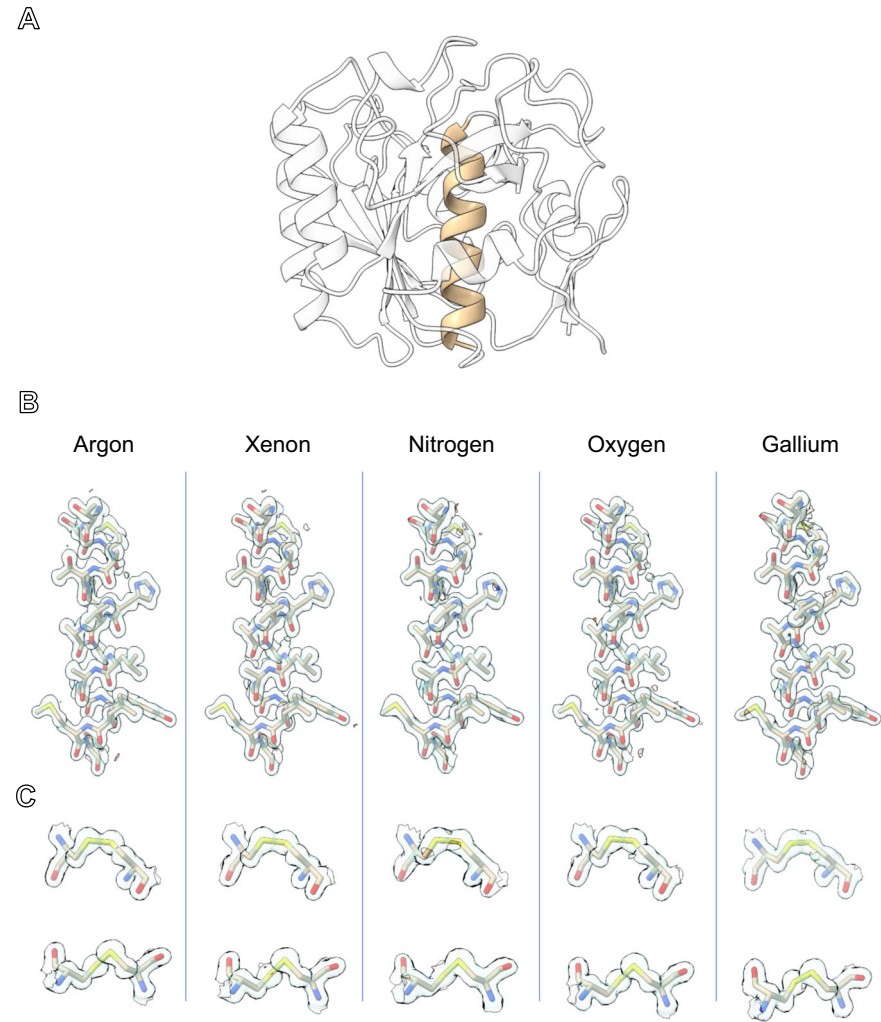

**Fig. 5 | The structure of proteinase K determined from plasma ion-beam milled lamellae. A** The structure of the serine protease, proteinase K, determined by MicroED from plasma ion-beam milled lamellae. **B** Maps for each plasma source and the prior best gallium structure from the same helix (residues 328–344) highlighted in **A. C** The two disulfide bonds in proteinase K (Cys$^{139}$–Cys$^{228}$ top, and Cys$^{283}$–Cys$^{364}$ bottom) for each structure. 2mF$_o$-DF$_c$ maps are all contoured at the 1.5 σ level, and the mFo-DFc difference maps are all contoured at ±3 σ level in green and red, respectively.

---

significantly better than previous results that required changing the phase of the LCP[11], and represents a clear path forward for the routine determination of GPCR crystal structures by MicroED.

## Discussion

We present a robust method to determine GPCR protein crystal structures by MicroED from unconverted LCP by targeting buried crystals using in-chamber cryo-CLEM, and plasma ion-beam milling. Using this approach, we determined the structure of the human adenosine receptor, A$_{2A}$AR, by MicroED. The protein was fluorescently labeled. Crystals were grown in LCP, looped and then smeared across an EM grid before freezing. The GPCR crystals were buried in dense LCP and could not possibly be identified by using FIB-SEM imaging. Instead, the crystals were located using an integrated fluorescent light microscope. We confirmed the viability of the method by using different dyes. Deep milling through LCP to the depth of the fluorescent crystals was accomplished using a plasma focused ion-beam rather than the traditional gallium beam. To achieve this result, we applied plasma focused ion-beam milling to thin cryogenically frozen biological material. To our knowledge, these are the first biological lamellae milled using plasma focused ion-beam sources for cryoEM experiments in a TEM. The speed of lamellae preparation indicates that xenon mills the fastest with the others following in the order of argon,

oxygen, and nitrogen. Although argon milled lamellae faster for the small proteinase crystals, this was because a higher current could be used without destroying these tiny crystals or visibly damaging them during the rough milling steps. For the A$_{2A}$AR crystals buried deep in LCP, much higher currents could be used. The qualitative metric of lamella cracking suggests that the highest rate of unbroken lamellae occurred using argon or xenon, whereas the lamellae that displayed the least curtaining would by either argon or oxygen. Crystallographic statistics show that the best data is obtained from either argon or xenon, with oxygen and nitrogen performing more poorly. Nitrogen milled lamellae were clear outliers as the worst of all categories overall. Although the oxygen milled lamellae showed better resolution and statistics, the structures from nitrogen or oxygen milled lamellae were of similar overall quality. The MicroED data collected from both argon and xenon milled lamellae of proteinase K were individually of better quality than any data previously recorded from gallium milled lamellae, indicating that there appears to be a clear improvement in data from lamellae prepared by these sources. The improved data quality may arise from reduced damage to the lamellae faces compared to gallium milled lamellae. An improved vacuum also prevents the rapid buildup of amorphous ice in the pFIB-SEM chamber. These improvements are in addition to the increased speed of preparing lamellae using a pFIB, where we manually prepared twenty lamellae across four

**Table 1 | MicroED structures of Proteinase K determined from plasma beam milled lamellae[a]**

| | Argon milled crystals | Xenon milled crystals | Nitrogen milled crystals | Oxygen milled crystals | Best merge |
|---|---|---|---|---|---|
| Data collection | | | | | |
| Space group | $P\,4_3\,2_1\,2$ | $P\,4_3\,2_1\,2$ | $P\,4_3\,2_1\,2$ | $P\,4_3\,2_1\,2$ | $P\,4_3\,2_1\,2$ |
| Cell dimensions | | | | | |
| $a, b, c$ (Å) | 67.02, 67.02, 107.53 | 67.05, 67.05, 107.02 | 67.12, 67.12, 106.87 | 67.26, 67.26, 106.81 | 67.02, 67.02, 107.53 |
| $\alpha, \beta, \gamma$ (°) | 90, 90, 90 | 90, 90, 90 | 90, 90, 90 | 90, 90, 90 | 90, 90, 90 |
| Resolution (Å) | 1.4 (1.45–1.4) | 1.45 (1.50–1.45) | 1.8 (1.86–1.8) | 1.5 (1.55–1.5) | 1.39 (1.44–1.39) |
| $R_{merge}$ | 0.3235 (1.784) | 0.2942 (1.475) | 0.5183 (1.798) | 0.759 (1.674) | 0.3396 (1.773) |
| $<I\,/\,\sigma I>$ | 7.61 (1.40) | 7.73 (1.62) | 4.18 (1.24) | 4.91 (1.03) | 10.78 (1.68) |
| Completeness (%) | 97.27 (96.17) | 98.61 (98.29) | 99.68 (99.21) | 96.20 (95.51) | 99.44 (97.05) |
| Redundancy | 25.8 (20.7) | 21.3 (15.5) | 15.2 (15.0) | 13.7 (11.0) | 54.8 (31.3) |
| Refinement | | | | | |
| Resolution (Å) | 1.4 | 1.45 | 1.8 | 1.5 | 1.39 |
| No. reflections | 47,738 | 43,468 | 23,288 | 38,542 | 49,781 |
| $R_{work}\,/\,R_{free}$ | 0.1374/0.1735 | 0.1387/0.1770 | 0.1679/0.2121 | 0.1634/0.2138 | 0.1252 / 0.1630 |
| No. atoms | | | | | |
| Protein | 2063 | 2052 | 2031 | 2047 | 2090 |
| Ligand/ion | 10 | 10 | 2 | 10 | 10 |
| Water | 307 | 294 | 237 | 344 | 284 |
| $B$-factors | 14.47 | 15.14 | 19.42 | 12.16 | 13.67 |
| Protein | 12.51 | 13.43 | 18.47 | 10 | 12.00 |
| Ligand/ion | 27.44 | 25.25 | 20.14 | 24.69 | 16.38 |
| Water | 27.21 | 26.76 | 27.49 | 24.63 | 25.92 |
| R.m.s. deviations | | | | | |
| Bond lengths (Å) | 0.015 | 0.009 | 0.004 | 0.002 | 0.014 |
| Bond angles (°) | 1.1 | 0.88 | 0.63 | 0.48 | 1.20 |

[a]Structures were derived from 5 crystals for argon and xenon, 4 for oxygen and nitrogen, and 12 for the best merge.

different sources twice as fast as what could be prepared using a gallium instrument.

Although this approach was successful for the $A_{2A}$ adenosine receptor, it may not be applicable to all membrane protein systems. This is simply because not every membrane protein could crystallize in LCP. Other crystallization methods such as vapor diffusion in the presence of detergents or lipid bicelles may be viable as described previously[8,42]. The ability to locate the membrane protein crystals buried in the LCP depends on the ability to label the sample with a fluorescent dye as autofluorescence or UV may not be sufficient[43]. Several tagging or labeling methods are available all of which may come with their own benefits and pitfalls. Label free methods, for example identification of nanocrystals with SONICC, provides a very sensitive tool but it does not work for all crystallization symmetries[44]. Regardless, further development using methods that do not rely on a dye could be beneficial.

We suspect that the MicroED data quality could be further improved by polishing the milled lamellae at lower accelerating voltages, as is the standard in materials applications[45]. The data collected here represents a first step into the application of plasma beam milling of biological samples for cryoEM investigations. Given the speed and quality of these initial results, we foresee application of this approach to automated lamellae preparation software with throughput gains of up to an order of magnitude over the current state-of-the-art. The improved resolution, data quality, and speed will correspond to improved signal-to-noise ratios in other cryoEM methods that prepare samples by FIB milling such as cryoET. Furthermore, the approach of plasma ion-beam milling buried membrane protein crystals identified using integrated fluorescence microscopy will accelerate the adoption of MicroED data collection from critically important membrane proteins.

## Methods

### Materials
Proteinase K was purchased from Sigma and used without further purification. Milli-Q water was used for all stock solutions. Cy3 fluorescent dye was purchased from Thermo Fisher and used without further purification. All stock solutions were membrane filtered three times. Tetraspecs fluorescent beads were purchased from Invitrogen.

### Protein purification
Expression and purification of $A_{2A}AR$, containing BRIL fusion protein in the third intracellular loop and a C-terminal truncation of residues 317 to 412 ($A_{2A}AR$-BRIL-$\Delta$C), were done as previously described[41,46].

### Growing protein microcrystals
Proteinase K was crystallized as described[47]. Protein powder was dissolved at a concentration of 40 mg/mL in 20 mM MES–NaOH pH 6.5. Crystals were formed by mixing a 1:1 ratio of protein solution and a precipitant solution composed of 0.5 M $NaNO_3$, 0.1 M $CaCl_2$, 0.1 M MES–NaOH pH 6.5 in a cold room at 4 °C. Microcrystals grew overnight.

The $A_{2A}AR$, protein was labeled on column with either Dylight550-NHS ester or Cy3-NHS ester in accordance with the FRAP-LCP protocol[48]. Labeling buffer contained 50 mM Hepes pH 7.2, 800 mM NaCl, 10% glycerol, 0.025% DDM 0.0025% CHS, 0.1 v/v% Cy3-NHS solution (4 mg/ml in DMF), 100 µM ZM241385. Labeling was carried out for 3 h at 4 °C. The excess of dye was washed off with the buffer without Dylight550 or Cy3-NHS ester. The sample was eluted in 3 cv of elution buffer (50 mM HEPES pH 7.5, 150 mM NaCl, 250 mM imidazole pH 7.5, 0.025%/0.005% (w/v) DDM/CHS, 10% glycerol, 100 µM ZM241385). The complex was concentrated to 30 mg/ml with Amicon

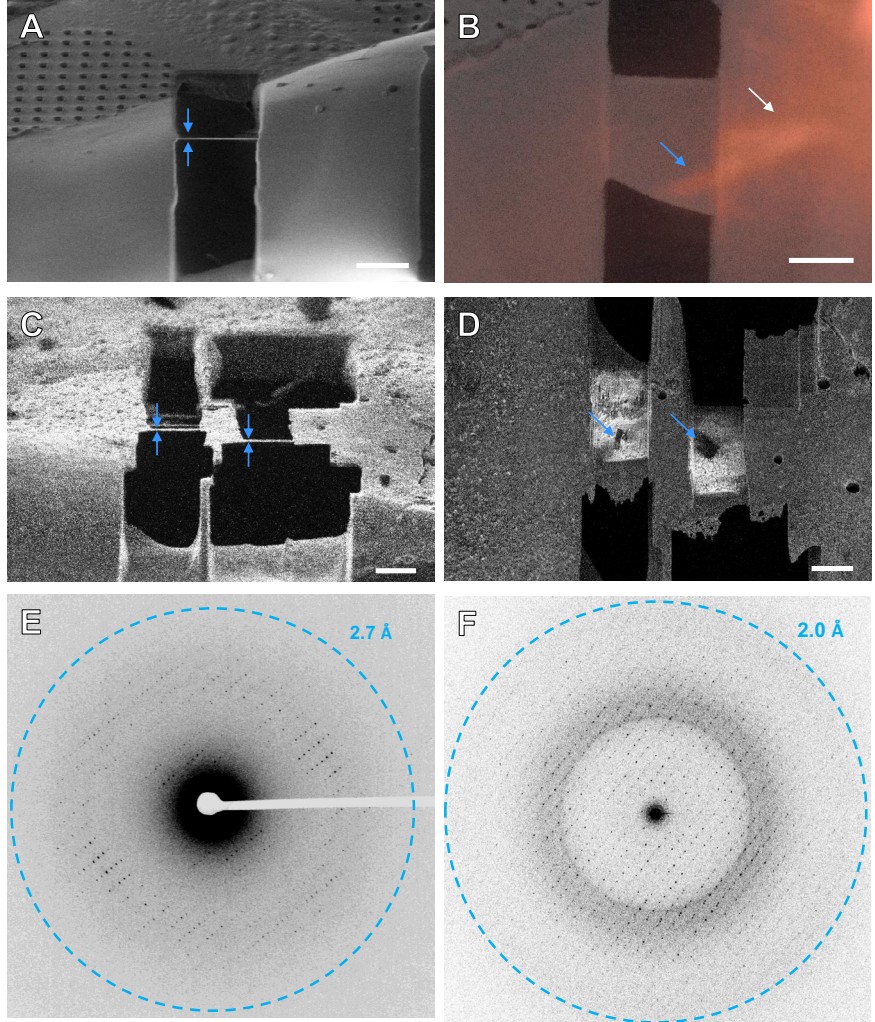

**Fig. 6 | Milling A$_{2A}$AR microcrystals in thick LCP and MicroED data collection. A** Final milled lamella of the GPCR crystal labeled by Cy3 in LCP viewed in the pFIB indicated by blue arrows. The final thickness is ~250 nm. **B** Overlaid SEM and iFLM 525 nm fluorescent images of the final lamella confirming the crystal survived the milling process. The blue arrow depicts a milled portion of the crystal, and the white arrow shows an unmilled area of the crystal identified by the fuzzier boundary. **C** Final milled lamellae of the GPCR crystals labeled by DyLight550 in LCP viewed in the pFIB indicated by blue arrows. **D** SEM image (iFLM overlay SI-Fig. 18) of the top of the lamellae in **C** showing clear dark regions where the crystals are located. **E** MicroED data from the lamella in **A** and **B**. **F** MicroED data collected from a lamella depicted in **C** and **D**. Scale bars 10 μm.

centrifugal filter units with 100 kDa molecular weight cutoff (Sigma Millipore).

A$_{2A}$AR-BRIL-ΔC in complex with ZM241385 were reconstituted into LCP by mixing with molten lipid (monoolein: cholesterol 9:1) using a syringe mixer in ratio 2:3.

Crystals for MicroED data collection were obtained in 96-well glass sandwich plates (Marienfeld). Precipitant solution contained 50–75 mM sodium thiocyanate, 100 mM sodium citrate pH 4.8, 28% (vol/vol) PEG 400, and 2% (vol/vol) 2,5-hexanediol. Crystals appeared within 24 h and reached full size within 7 days.

### Grid preparation

**Proteinase K grids.** Quantifoil Cu 200 R 2/2 holey carbon TEM grids were glow-discharged for 30 s at 15 mA on the negative setting immediately before use. These grids were loaded into a Leica GP2 vitrification robot. The robot sample chamber was loaded with filter paper and set to 4 °C and 95% humidity for 1 h before use. In all, 3 μL of protein crystals from the center of the proteinase K tubes were applied to the carbon side of the glow-discharged grid and allowed to incubate for 10 s. Grids were then gently blotted from the back for 10 s. These grids were then immediately plunged into super-cooled liquid ethane. Grids were stored in liquid nitrogen until use.

**FIB-SEM/iFLM depth calibration grids.** In all, 4 μm Tetraspecs fluorescents beads (Invitrogen # T7284C) were diluted to a ratio of 1:10 in a 50% glycerol aqueous solution. 3 μL were pipetted on glow-discharged Quantifoil grids Cu R2/2 (EMS # LFH2100CR2) that were then manually back blotted ~1–2 s and plunge frozen in liquid nitrogen. Alternatively, 0.3–0.5 μL droplets were deposited on the grids and the grids were frozen without prior back blotting in order to create domes of glycerol with the Tetraspecs incased in them. Grids were clipped and loaded in the FIB-SEM.

**A$_{2A}$AR grids.** Crystals were looped from glass sandwich plates using a 100 μm MiTeGen dual thickness micro mount and carefully transferred to glow-discharged Cu200 R2/2 grids that had been pre-clipped. Looping was conducted under a light microscope next to a humidifier to prevent the LCP from drying out and changing phase during the transfer. Loops full of LCP and crystals were gently slid across the surface of the grid, and the grids were then immediately plunged into

**Table 2 | MicroED structure of A$_{2A}$AR determined from microcrystals in LCP[a]**

|  | A$_{2A}$AR |
|---|---|
| Data collection |  |
| Space group | C 2 2 2$_1$ |
| Cell dimensions |  |
| a, b, c (Å) | 39.08, 177.51, 139.29 |
| α, β, γ (°) | 90, 90, 90 |
| Resolution (Å) | 2.0 (2.07–2.0) |
| R$_{merge}$ | 0.3251 (0.464) |
| <I / σI> | 4.31 (1.29) |
| Completeness (%) | 72.49 (16.12) |
| Redundancy | 4.7 (1.7) |
| Refinement |  |
| Resolution (Å) | 2.0 |
| No. reflections | 24,440 |
| R$_{work}$ / R$_{free}$ | 0.2202/ 0.2792 |
| No. atoms |  |
| Protein | 3012 |
| Ligand/ion | 1070 |
| Water | 174 |
| B-factors | 23.1 |
| Protein | 22.62 |
| Ligand/ion | 28.81 |
| Water | 16.99 |
| R.m.s. deviations |  |
| Bond lengths (Å) | 0.027 |
| Bond angles (°) | 0.91 |

[a]A$_{2A}$AR was derived from 3 crystal lamellae – 2 collected on a Titan Krios using a Falcon 4 detector, and 1 using a Talos Arctica using a CetaD detector.

liquid nitrogen. All grids were stored at liquid nitrogen temperature before further experiments.

**Calibrating the milling depth between iFLM and pFIB images**
We first used the in-chamber fluorescence light microscopy (iFLM) to localize a thick area containing numerous Tetraspecs at various depths. We milled off the top of the glycerol pile to create a small surface visible in light microscopy, defining our "zero-depth" reference. We registered the depth of the Tetraspecs relative to our zero-depth reference. Then, alternatively milling at high currents by increments of 10–4 μm deep and monitoring the disappearance of the Tetraspecs by iFLM allowed us to track the milling depth at which each Tetraspec disappears and compared it to the iFLM-measured depth.

To estimate the depth and then monitor the disappearance of the fluorescent Tetraspecs in the glycerol, fluorescent stacks and reflection stacks were acquired using the iFLM setup in our Hydra: a fixed ×20 objective, 0.7 numerical aperture, and working distance of 0.6 mm.

Fluorescence was used to track the beads and reflection was used to monitor the topology of the milling area, mainly to accurately determine the surface of the milled area.

The light source is a 4 LED system (385, 470, 565, and 625 nm) (Thorlabs LED4D242). For fluorescence imaging a fluorescent quad-band filter cube (Semrock LED-DA/FI/TR/Cy5-B-000) is introduced on the light path. For reflective imaging, an empty filter cube is introduced in the light path. The detector is a 3088 × 2064 frame, with a physical pixel size of 2.4 μm. With the ×20 objective, the pixel size is 120 nm.

Stacks of the Tetraspecs were systematically acquire using the fluorescent (excitation wavelength of 470 nm) and reflection modalities with the same parameters and a shuttle inclination of 25° resulting in an image normal to the plane of the EM grid. Recorded data were binned by 2, with 100% intensity excitation and 1 or 5 ms exposure for each optical slice for the fluorescence and reflection mode, respectively. The Z-step was consistently set to 2 μm.

To mill through the thick glycerol piles, milling was performed with the xenon beam at 30 kV – 15 and 60 nA, at a shuttle inclination of 0° (resulting in a grazing milling angle of 11°) in order to mimic real milling conditions. Milling box X and Z dimensions depended on the size of the glycerol pile. The Y-dimension, defining the milling step, ranged from 20 to 4 μm (when getting closer to the bead positions). SEM and FIB imaging was done at 500 V–25 pA and 30 kV–3 pA, respectively. Both used the Everhart-Thornley Detector (ETD).

Fluorescence and reflection stacks were combined into multi-channel stacks using FIJI[49]. To overlay the two images, a maximal intensity projection was calculated to see all the Tetraspecs present in the stack at once. The first stack acquired was defined as our zero-depth reference and used to estimate the depth of the Tetraspecs encased in the glycerol. The subsequent stacks generated in between each milling increment are then processed the same way and compared to the previous stack. When a Tetraspec disappears between two iFLM stacks associated to a milled depth, the disappearance depth considered was defined as:

$$disapearance\ depth = \lceil Previous\ milled\ depth\rceil \\ + \frac{\lceil Current\ milled\ depth\rceil - \lceil Previous\ milled\ depth\rceil}{2} \quad (1)$$

Exceptions were made when it appeared that some of the Tetraspecs were half milled, indicating that the current milled depth is spot on the bead, and therefore the current milled depth was used as the disappearance depth.

Plotting of the Tetraspec depth iFLM estimation vs disappearance depth was done using R and R studio and the following packages: *tidyverse* and *here*. As a reference, we included the theoretical FIB-view depth, which is a function of the milling angle relative to the grid plane. In a perfect system where iFLM and milling depth measurements are accurate, the Tetraspecs should disappear at the projected FIB-view depth, defined as:

$$Projected\ FIB\ view\ depth = \lceil iFLM\ depth\ measurement\rceil\ x\cos(a) \quad (2)$$

Where $a$ is the milling angle – 11° in this work.

**Protecting vitrified biological samples from the plasma beam**
Vitrified biological samples need to be protected from the ion-beam during imaging and milling. Inside the FIB-SEM, the sample is typically coated by either a thin layer of pure platinum grains using a sputter coater, a volatile hydrocarbon platinum mixture[50] using a gas injection system (GIS), or a combination of both[13,14,33]. The sputter-coated platinum layers serve to make the sample conductive and reduce charging artifacts, whereas the thicker GIS deposited platinum protects the sample from the ion-beam during imaging and milling. During these experiments, images are taken using the ion-beam at the lowest current to monitor the sample thickness and adjust for drift or sample movement.

Even at low flux, the ion-beam can damage the sample during imaging[18]. Milling is typically conducted at much higher beam currents than imaging[12,14,23]. For this purpose, it has become a routine to protect the samples by coating the specimens in a thick layer of platinum using a GIS. For many samples, such as mammalian cells that are relatively

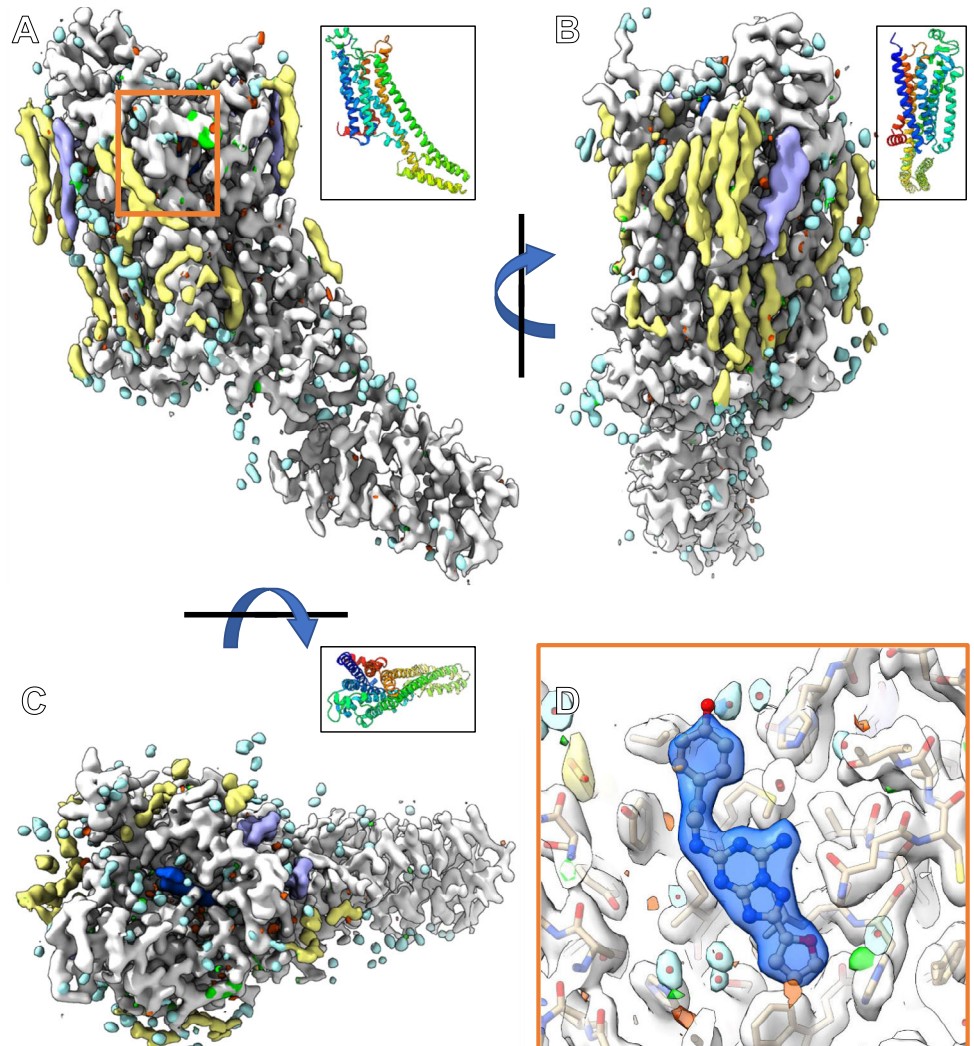

**Fig. 7 | The 2.0 Å MicroED structure of A$_{2A}$AR from microcrystals buried in thick LCP. A–C** Showing the final 2mFo-DFc density and mFo-DFc difference maps contoured at 1σ and ±3 σ at various orientations. The corresponding ribbon diagram is showing in rainbow in the top right corner for each density. **D** The binding pocket showing the density and difference densities around the bound antagonist located in the orange square in **A**. All contours appear at identical levels. The 2mFo-DFc density is colored white around the main chain, blue around the antagonist, purple around cholesterols, yellow around oleic acids, and light blue/teal around water molecules.

flat, this approach gives a reasonably smooth, protective layer. For samples with more challenging aspect ratios, such as crystals, the GIS deposition coats differentially because of the facets of the crystals shadowing the grid differently. GIS deposition on grids with crystals often leads to platinum layers that are not homogenous with bubbles, and imperfections.

Prior MicroED investigations of thinned crystals all used a gallium focused ion-beam[10,17,18]. For many of these samples, a sputter coating of platinum was sufficient to protect the crystals by merely increasing the sputtering time and thereby thickening the layer (60–180 s, ~10–100 nm of platinum). This approach worked because crystals were milled using very low currents (maximum of typically 300 pA) and the gallium beam size was small enough to overlap very little with the exposed lamella face. This pFIB instrument is not equipped with a sputter coater. Therefore, a different strategy for GIS platinum coating needed to be developed.

At room temperature, the ion or electron beam is used to chemically cleave the volatile mixture, applying material only to the imaged area of the sample[51]. Under cryogenic conditions, the GIS platinum sticks to the grid due to the temperature difference between the volatile, carbon-rich platinum and the cryogenic sample. The rate

of platinum deposition at cryogenic temperatures by the GIS is typically too fast to be modified by adding exposure from either the electron or ion beams. After multiple trials, we were able to generate a consistent, dense platinum layer fully protecting all of the crystals along the milling direction. This was accomplished by moving the sample further from the GIS needle to slow the deposition rate and simultaneously imaging the whole grid with a low accelerating voltage, high-current xenon beam (Supplementary Fig. 4). This coating scheme typically doubled the success rate of lamellae preparation in our hands compared to any conventional method of GIS deposition prior to PFIB milling of these crystals.

Whole grid atlases/montages were created from tiles of individual images taken by the scanning electron microscope operating at an accelerating voltage of either 500 V or 2 kV and beam current of 13 pA in the MAPS software (Thermo Fisher). From the montages, crystals were selected that were not within a few μm of a grid bar, nor within 3 grid squares of the edge of the grid. Dozens of crystals across over ten grids were identified in this way to test various platinum protection setups and testing of milling strategies. After many failures, the results of the beam comparison within this investigation were conducted, where 20 crystals were identified under these criteria.

## Machining proteinase K microcrystals using the pFIB

The twenty crystals were divided into four groups of five, one group for each plasma ion-beam to test. For each gas source, crystals were milled sequentially using identical milling strategies—the same preset pattern, beam currents, and times. The Z-depth of the patterns and currents were the only parameters adjusted. For each crystal and gas source, the milling was conducted in four steps with approximately the same parameters summarized in Supplementary Table 2. Generally, all milling was conducted using the cleaning cross section for each pattern with 85% overlap between both X and Y spot positions. The first milling step used an ~1 nA current to mill two boxes of $6 \times 6$ μm separated in the middle by 2 μm. The second utilized two cleaning cross sections of $5 \times 1$ μm in size separated by 1μm that used an ~0.3 nA current. The third step consisted of two $5 \times 0.5$ μm boxes separated by 0.5 μm with a beam current of 0.1 nA. The final milling step consisted of two $5 \times 0.3$ μm boxes separated by 300 nm that were milled with a beam current of ~30 pA. Currents between sources were chosen to be within 1 aperture number from the prior source to minimize the number of alignments between experiments (Supplementary Table 2). All cleaning cross sections above the lamella were milled from the top to the bottom, whereas all the cross sections below the lamella were milled from the bottom to the top. The sputtering rate for a drawn pattern in the microscope software is set to solid silicon, which is much denser than vitrified water or biological materials. We empirically determined reasonably adapted milling times by varying the dictated Z dimension, or depth of the drawn patterns. For xenon, we used a depth setting of 5, 3, 2, and 2 μm deep for each step. For argon, these were 6, 3, 2, 2 μm deep. For nitrogen, we used 20, 10, 4, 4 μm. Finally, we used 8, 6, 4, 4 μm for the oxygen beam. These settings are summarized in Supplementary Table 1. In most cases, these were higher than strictly necessary to ensure second passes would not be needed. However, even with the depths used, the nitrogen lamellae required constant manual intervention that was still unable to rescue some of the lamellae. Typically, argon and xenon lamellae were completed with total milling times of between 4 and 20 mins depending on alignments between milling steps and various manual microscope operations. Each nitrogen lamella took ~15–30 mins of on-sample milling time. For oxygen, this was similarly 15–30 mins per lamella. A complication to the timing was the manual operation and shortcomings of specific gasses. For example, focusing the argon and xenon beams is more challenging than a gallium beam, but straightforward. The oxygen and nitrogen beams are very difficult to focus and align at low beam currents. Positioning lamellae was also much easier for the heavier ions since the focused images were much sharper in general. Finally, imaging lamellae using the various ion beams changes the contrast in the electron beam due to the differential breakdown of the GIS-deposited platinum over time and differing by each ion. For example, oxygen lamella #2 (Supplementary Fig. 8) was all but invisible after milling, and even after repeated attempts, the SEM image had to be zoomed out to locate the lamella. In our experiments, the contrast changing of the GIS deposited platinum without the ion-assisted deposition described herein was much worse, essentially making many attempts at milling with nitrogen or oxygen much more challenging than simply using a gallium beam source.

**Identification and machining of A$_{2A}$AR crystals.** Frozen A$_{2A}$AR grids were transferred into the pFIB/SEM under cryogenic conditions. All-grid montages were collected using the SEM operating at 500 V prior to platinum coating. The low accelerating voltage prevents damaging the sample and enables visualization of the sample with improved contrast compared to the platinum coated sample. After coating, almost all images in the SEM appear similar. Areas of thick LCP seen in the SEM were inspected in the iFLM using 385/470/565 nm light images. When an area of interest was located, i.e. with potential crystal targets located more-or-less in the center of a grid square, the grid was

presented at a normal incidence angle (90°) to the FIB beam. Holes were milled in the sample using the "Spot Burn" at locations visible in both the SEM at normal incidence and FIB view at grazing-incidence. Determining optimal positions of the milled fiducial markers ("FIBucial markers") required going back and forth between normal and grazing incidence FIB views. The fiducials were milled with the argon beam at 30 kV, 7.6 nA and for a duration of ~15 s (Supplementary Fig. 17A). An overview image of the area was acquired for further correlation at 60 pA (for argon) or 30 pA (for Xe) at normal and grazing incidence using the FIB beam (Supplementary Fig. 17A, B), and a fluorescent stack was acquired at normal incidence with the iFLM (Supplementary Fig. 17C).

The fluorescent stack was deconvoluted using the DeconvolutionLab2 FIJI plugin[52]. Experimental PSFs were determined on our system with 100 nm TetraSpecs fiducial markers (Invitrogen # T7284C) and the Huygens software (https://svi.nl/Huygens-Software).

The deconvoluted stack was imported in 3DCT[53], resliced into isotropic pixels (240 nm × 240 nm × 240 nm instead of 240 nm × 240 nm × 1000 nm). The milled fiducial markers and crystals of interest were designated in 3D (Supplementary Fig. 17C, pointed at in green). The Z position of each milled marker was assumed to be where the hole had sharpest features by eye. FIB views were imported in 3DCT and the milled markers were designated in 2D. Designation of at least five markers in both modalities allowed correlations with a residual mean standard error less than 5 pixels, and typically <3 pixels (Supplementary Fig. 18A, C).

The resulting correlated images show the FIB view with the predicted positions of the milled fiducial markers in green and the designated crystals in red (Supplementary Figs. 18A, D, E). This allowed accurate positioning of the lamella. Further FIB views later in the milling process were quickly correlated against the same fluorescent stack, as long as the same fiducials were visible. Between each step of milling, single iFLM images at the known Z position allowed us to monitor the crystal.

This fluorescence-guided method was used to create a lamellae ~5 μm thick. At this point, the Z-resolution of our system did not allow precise correlation. From here, milling was conducted from the top in increments of 500 nm until we reached the crystal. In between each milling, the sample was imaged by iFLM and the crystal was checked for changes in its shape and increases in the fluorescence intensity, which would indicate that the crystal is exposed at the surface of the lamella. The sample was then screened by SEM at low current (1.2 kV at 13 pA) in immersion mode. This allowed to see contrast differences at the surface of the lamella between the crystal and the surrounding LCP (Supplementary Fig. 18D, F, G). This was iteratively applied until the final thickness of 300 nm was reached. A final polishing step at the surface of the lamella was applied before immediate transfer to the Titan Krios. The milling parameters used for the A$_{2A}$AR crystals are given in Supplementary Table 2.

## MicroED data collection

Grids containing milled proteinase K crystals were rotated such that the TEM rotation axis was 90° from the plasma-beam milling axis. The grids were then loaded into a cryogenically cooled Thermo Fisher Titan Krios 3Gi transmission electron microscope operating at an accelerating voltage of 300 kV. Low magnification montages of each grid were collected at a magnification of 64× and used to locate the milled lamellae. Each lamella was brought to its eucentric position before data collection. MicroED data were collected by continuously rotating the stage at a rate of ~0.15°/s for 420 s, covering a total rotation range of ~63°, respectively. This typically spanned the real space wedge corresponding to approximately −31.5° to +31.5°. Data were collected using a 50 μm C2 aperture, a spot size of 11, and a beam diameter of 20 um (SA aperture). Under these conditions, the total exposure to each crystal was ~1.0 e⁻ Å⁻². Diffraction data were collected

from a small, isolated area from the middle of each lamella of ~2 μm in diameter using the 100 μm selected area aperture to remove unwanted background noise. All data were collected using twofold binning and internally summed such that each image recorded a 0.5 s exposure spanning ~0.075° of rotation. In this way, each image stack contained 840 images, the last of which was discarded for having an unequal number of frames. A single sweep of continuous rotation MicroED data was collected from each lamella.

For $A_{2A}AR$, one MicroED dataset was collected on a Talos Arctica operating at liquid nitrogen temperatures at an accelerating voltage of 200 kV. Data were collected by continuously rotating at a rate of 0.5 °/s for 160 s, spanning a real space wedge from −40° to +40°. Data were collected on a CetaD CMOS 4096×4096 detector operating in rolling shutter mode with correlated double sampling active. Two additional datasets were collected using the Falcon 4 in electron counting mode.

### MicroED data processing
Movies in MRC format were converted to SMV format using a parallelized version of the MicroED tools (https://cryoem.ucla.edu/downloads). Each proteinase K and $A_{2A}AR$ dataset was indexed and integrated using XDS. All datasets were scaled using XSCALE. The final $A_{2A}AR$ data was produced by merging 1 dataset collected on the Talos Arctica using a Ceta-D and 2 datasets collected on a Titan Krios using a Falcon 4. For all crystals, the space group was verified using POINTLESS. Data were merged without scaling using AIMLESS, the subsequent intensities were converted to amplitudes in CTRUNCATE, and a 5% fraction of the reflections were assigned to a free set using FREERFLAG[54].

In order to achieve the best model possible from our collected data and to test if data derived from different ion sources could reasonably be merged together, we created an additional merged data set from across all the lamellae. First, a naïve merge of all the integrated datasets was conducted. To identify which datasets from each source merged best, isoclustering[55] was performed. Poorly contributing data were discarded and the remaining datasets were automatically assigned weights and merging order to yield a "Best Merge" from this subset (Fig. 3 and Table 1). This merged dataset was composed of 12 of the 20 individual datasets – 5 argon datasets, 5 xenon datasets, and 2 oxygen datasets. This final merged dataset had overall statistics superior to any of the individual datasets or subsets of merged data from individual gases and to a slightly better resolution (Table 1). The structure of proteinase K was determined from this dataset and refined identically to the other sources. The refined structure from this merged data had an overall R work and R free of 11.92/16.34. These statistics were better than the models derived from argon and xenon alone. The results suggest that merging data from across different ion sources is possible without degradation of the model. It could be that there is some benefit in merging data between the sources given the improved metrics, however it is difficult to separate the improvements in statistics and resolution from the increase in multiplicity.

### Structure solution and refinement
The structures of proteinase K were determined by molecular replacement in PHASER using the search model 6cl7. The structure of $A_{2A}AR$ was determined by molecular replacement using 4EIY as a search model. The solutions were refined in phenix.refine. For proteinase K models, the first refinements used isotropic B-factors and automatic water picking that resulted in $R_{work}$ / $R_{free}$ of ~0.18/0.20. The refined model was inspected in Coot. Several calcium and $NO_3$ ions were placed in the difference maps, an incorrectly assigned residue (SER[312] instead of ASP[312]) was fixed, and alternative conformations were identified for several residues. Occupancies were refined for nitrate ions and alternate side chain conformations. This model was refined again in Phenix using the same settings that resulted in approximate

$R_{work}/R_{free}$ of 0.16/0.19. After another visual inspection in Coot, the model was refined again in Phenix using automatic water picking and anisotropic B-factor refinement for all atoms that resulted in $R_{work}/R_{free}$ of 0.15/0.18. From here, the model was refined again in REFMAC5 using automatic matrix weights, anisotropic B-factors, and added hydrogens, where the final $R_{work}/R_{free}$ dropped to 0.12/0.16. The $A_{2A}AR$ model was iteratively refined in phenix.refine and inspected in COOT to a final $R_{work}$ / $R_{free}$ of 22/27 and resolution of 2.0 Å.

### Figure and table preparation
Figures were prepared using ChimeraX[56], FIJI[49], the matplotlib package in Python 3.6 in a Jupyter notebook and R. Figures were arranged in PowerPoint and Adobe Illustrator, and Tables were arranged in Excel. Maximum intensity projections were calculated in FIJI[49]. The manuscript was written in Word 2016 (Microsoft).

### Statistics and reproducibility
In this work, all the proteinase lamellae were made on a single grid originating from the same batch condition. Twenty total crystals were selected that were located in ideal positions on the grid. Each lamella preparation was repeated five times ($n = 5$) using the same protocol. All individual crystals, finished lamellae, and diffraction are presented in the Supplementary Information without omission. The $A_{2A}AR$ lamellae presented originated from two different grids prepared using two different fluorophores and protein expression steps. One lamellae was made from the first condition, and two were made from the second. All three lamellae diffracted, though to different resolutions and qualities.

### Reporting summary
Further information on research design is available in the Nature Portfolio Reporting Summary linked to this article.

## Data availability
The data that support this study are available from the corresponding authors upon request. The EM maps have been deposited in the Electron Microscopy Data Bank (EMDB) under accession codes EMD-29588 (Proteinase K, xenon); EMD-29590 (Proteinase K, argon); EMD-29596 (Proteinase K, nitrogen); EMD-29595 (Proteinase K, oxygen); EMD-29587 (Proteinase K, best merges), and EMD-29586 ($A_{2A}$-AR). Coordinates have been deposited in the Protein Data Bank (PDB) under accession codes 8FYP (Proteinase K, xenon); 8FYQ (Proteinase K, argon); 8FYS (Proteinase K, nitrogen); 8FYR (Proteinase K, oxygen); and 8FYN ($A_{2A}$-AR).

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

## Acknowledgements

This study was supported by the National Institutes of Health P41GM136508 and the Department of Defense HDTRA1-21-1-0004. The Gonen laboratory is supported by funds from the Howard Hughes Medical Institute. We would like to thank Abhay Kotecha and Ron Kelley from Thermo Fisher Scientific for useful discussions.

## Author contributions

T.G. directed the research. M.W.M., A.S., M.T.B.C., W.J.N., and S.J.W. prepared samples. M.W.M., A.S., M.T.B.C., W.J.N., and S.J.W. collected the data. M.W.M., M.T.B.C., and J.H. processed the data. The manuscript was written by M.W.M. and T.G. with input from all authors. T.G. and M.W.M. conceived the project.

## Competing interests

The authors declare no competing interests.
