## [Peer Review File · Nature Communications]

A robust approach for MicroED sample preparation of lipidic cubic phase embedded membrane protein crystalsReviewers' Comments:

Reviewer #1:

In this manuscript, the authors detail methods to produce higher quality lamellae for structure determination using microED while also providing additional technological developments for solving structures of integral membrane proteins embedded in unconverted lipidic cubic phase. The careful analysis of the different gases that can be used for milling is thoroughly executed and the very careful considerations for each type of gas, operations, and expectations are very well defined. The current limitations of solving structures using gallium and current microED technologies and the need for plasmas is carefully outlined and reasonably sound (caveat outlined below). The manuscript is well-written but there are some sections, including the Outlook section, that are choppy compared to other sections of the manuscript and could benefit from additional smoothing.

Overall, this reviewer has only minor considerations outlined below. The figures are clear and data is of high quality. However, the expectation from the title does not match with the article in its entirety. Indeed, a “robust” method for imaging crystals in LCP with only one detailed success is a bit of a misnomer. Including additional datasets of A_{2A}AR or perhaps a different IMP would bolster the claims of the manuscript. Of particular concern is the requirement of fluorescently labeling the protein prior to crystallization and whether this can be broadly applicable given the sensitivity of crystallogensis.

Specific comments:

-“A fundamental issue with attempting ion-beam milling of LCP embedded crystals is that this material is exceptionally difficult to mill using a gallium beam. Typically, the LCP will begin to indent, turn black, and then deform rather than being removed from the sample. Milling under these conditions is essentially impossible and has prevented milling into thicker LCP areas on the grids. Finding a method to mill away LCP without changing the phase requires a new approach to milling thick samples that does not involve a standard gallium ion beam.” (page 2)

There is no reference associated with this statement nor images within this manuscript. Given the technical nature associated with developing plasmas to ameliorate this observation, even if anecdotally well documented in the field, the results outlined herein would benefit from direct comparisons.

-The authors detail methods using fluorescently-labeled protein in LCP. However, this may not always be applicable and could be seen as a critical limitation for such a robust approach. Notably, discussion pertaining to this potential limitation is lacking in the manuscript. What if the protein cannot be labeled prior to crystallization? Would small molecule crystal soaks suffice? Is tryptophan fluorescence insufficient due to poor quantum yield? Are there alternative methods that can be explored? What are the size limitations of the crystals that can be observed using this method?

-Figure 3- I think it would benefit to show exemplar images from the SI in the main figure since so much discussion pertains to the quality of lamellae produced and their impacts on final data and structure quality.

-Figure 5- This figure would greatly benefit from images of the density for different regions of A_{2A}AR with their regions in panel D highlighted (like Figure 4A). At 2.7 Å resolution the quality of side chain density can vary widely. Alone, panel D does not provide much information besides what a GPCR looks like (which is common knowledge at this point).

-Not of much significance but was the A_{2A}AR structure was determined on an Arctica compared to a G3i for the proteinase K structures out of convenience (likely)? Or other reasons that are worth discussing?

Reviewer #2:

Remarks to the Author:

This manuscript by Martynowycz and coworkers describes methods for preparing protein microcrystalline lamellae for subsequent investigation by MicroED, for the ultimate goal of determining structures from membrane protein crystals grown in LCP, particularly of GPCRs. To attain this goal, in addition to the milling technique, the authors also developed procedures to:

- Test available plasma ion species for optimal milling and imaging conditions with proteinase K
- Use an integrated fluorescence microscope to find labeled protein crystals of interest for FIB/SEM
- Correlate the fluorescence images with SEM images for precise milling
- Determine the proper depth for crystal targeting
- Protect the lamella from ion beam damage

The manuscript clearly explains the methods involved and rationale behind each step. It combines previous ideas (e.g., trace labeling of protein with fluorescent dye, milling of frozen-hydrated biological material, etc.) and builds upon them using new technology (using a plasma focused ion beam/scanning electron microscope, pFIB/SEM).

As multi-ion species dual-beam FIB/SEM instruments have only recently become commercially available, to my knowledge this manuscript is the first to feature pFIB/SEM milling on biological material, in particular for subsequent cryo-EM investigation (by transmission electron microscopy, TEM). Therefore, lessons learned from this work may guide the location and milling of fluorescently-labeled biological targets (e.g., cells, tissues, etc.) for subsequent investigation by cryo-electron tomography (cryo-ET). In addition, this work expands the utility of the LCP method for crystallographic structure determination, as it provides another technique for diffraction data collection in addition to serial femtosecond crystallography (SFX).

Comments:

1 – Line 106 mentions that “higher Z sources sputter material faster and damage the surfaces less.” A detailed comparison of data quality from different plasma sources was done to assess diffraction quality and crystallographic statistics via MicroED, and showed that argon, followed by xenon, produced the best results. It would be interesting to compare SRIM simulations of different plasma beam sources on 200-300 nm-thick protein crystal lamellae with the experimental results, as well as with the previous report of SRIM simulations on tungsten metal, to guide further characterization of the ideal pFIB-SEM setup.

2 – There are other ways to detect protein in crystals, for example, UV imaging and second order nonlinear imaging of chiral crystals (SONICC), even though those may not be as specific as fluorescent labeling. Are these methods viable as possible alternatives for LCP protein crystal detection in a pFIB/SEM machine?

3 – The paragraph beginning on line 301 describes MicroED data collection, processing, and structural analysis of the GPCR lamella, and makes a comparison with their previous structure reported in Martynowycz et al. 2021 (PNAS). The refined density map of the A2AAR structure is not shown or uploaded as a supplement in this study, though it was described in the text. It would be helpful to the readers to see this data, particularly the refined density map, for comparison with the previous structure (of data collected after conversion to the sponge phase), and also since density maps were shown of the proteinase K data in this work (Figure 4).

4 – Please address why the A2AAR dataset was indexed in DIALS vs XDS (line 566), since XDS worked for indexing proteinase K. XDS could not find the indexing solution?

Reviewer #3:
Remarks to the Author:
Please see attached PDF file.

In this manuscript, the author applied plasma focused ion beam (pFIB) technique to prepare thin crystal lamellae of protein microcrystals. They first systematically studied the effects of different plasma sources on the quality of proteinase K crystal lamellae, as well as the quality of subsequent MicroED data collected on these lamellae. The structure of proteinase K was then refined against each MicroED dataset as well as the best merged dataset in order to identify the best ion source for pFIB milling. The authors then chose to use the most suitable ion source, xenon, to prepare a lamella of A_{2A}AR crystal embedded in LCP without transferring the LCP to the sponge phase (Martynowycz et al. *PNAS* **2021**). They employed a novel approach using an integrated fluorescence light microscope (iFLM) inside of a FIB/SEM to identify fluorescently labeled crystals embedded deep in a thick LCP layer. One thin crystal lamella was then prepared by xenon pFIB. The structure of A_{2A}AR was then solved using MicroED data collected on the lamella.

The authors made two major claims in the manuscript:

1. Crystallographic data processing and structure refinement against data collected on argon and xenon milled crystals resulted 'significantly better' quality 'than any prior investigation of proteinase K by MicroED'.

This claim is not exactly true as can be seen in Table R1. The authors recently reported that by using FIB milling with Gallium and collecting MicroED data on a direct electron counting detector (Falcon 4i), it was possible to obtain high quality MicroED data from proteinase K crystal lamella (Martynowycz *Nat. Methods* **2022**). Data statistics and key structure refinement indicators reported in this work show no significant improvements over the previous reported work. The small differences in R_{work} and R_{free} values are probably due to other reasons such as the multiplicity of the data. Moreover, when studying the structure models and maps obtained from lamella prepared by each plasma source and the prior best gallium structure (Martynowycz *Nat. Methods* **2022**) shown as figure 4 in the manuscript, I could not see huge differences between them, apart from maybe the slight resolution reduction of the map in the Nitrogen case.

Nevertheless, one of the benefits of using pFIB milling seems to be the faster milling speed, the ability to mill through thicker sample, and lower implantation depth.

2. '... The LCP needs to either be converted to the sponge phase or entirely removed from the path of the ion-beam to allow identification and milling of these crystals. Unfortunately, conversion of the LCP to sponge phase can also deteriorate the sample. Methods that avoid LCP conversion are needed.'

I agree that conversion of the LCP to sponge phase may deteriorate embedded crystals in some cases, but the example provided by the authors do not support this argument. Table R2 summarizes MicroED data processing and structure refinement statistics of A_{2A}AR reported in this work and in Martynowycz et al *PNAS* **2021**. There is no obvious difference between data collect on lamella milled from crystal embedded in LCP layer directly and from crystal embedded in sponge phase layers. An additional example where the authors could clearly

show the advantages of milling directly from LCP is needed to support their claim. Furthermore, milling crystals embedded in thick LCP layers is technically demanding and less available comparing to conventional cryo-FIB by Gallium ion.

Based on the above arguments, I do not support the publication of this manuscript in Nature Communications. I do however think this work is technically sound, and the results are well presented. I appreciate the systematic studies conducted by the authors. I hope after revising the manuscript, the manuscript can be published in a more specialized journal.

In addition, I would like to address some minor concerns:

- Why did the authors test the effects of different ion sources using crystals embedded in vitrified ice, if they want to study direct pFIB milling of crystals in LCP (Sup Figure 2)? Why didn't they test different ion sources using for example proteinase K crystals embedded in LCP layers? It would provide direct evidence on how each source would perform on LCP milling. It will also be more reasonable to find the best milling conditions using such sample.
- Page 2, Line 85-87, some early work on FIB milled lamella for MicroED were not cited: Duyvesteyn et al. *PNAS* **2018** (first report on FIB milled crystal lamella for MicroED), Li et al. *Biophysics Reports* **2018** (doi: 10.1007/s41048-018-0075-x) Zhou et al. *Journal of Structural Biology* **2019** (doi: 10.1016/j.jsb.2019.02.004)
- Page 3, line 109-110, the improved signal-to-noise ratio is probably attributed to the increased multiplicity.
- Page 3, Line 115, I would not call the iFLM imaging on-the-fly. It is more like switching between pFIB milling and iFLM inspection. (also on page 6 line 290-291, page 7 line 350-351). On-the-fly means imaging while milling.
- Page 3, Line 148-149, will tagging protein with a fluorophore prior to crystallization affect the crystallization process in some cases? Could fluorophore tags be added to microcrystals instead?
- Page 4, Line 159, what is the accuracy of the overlay? I would assume for milling microcrystals, it should have sub-micrometer accuracy.
- Page 5, Line 212, are there any evidences that 'ice contaminations and breakage not observed in the SEM prior to loading in the TEM are attributed to the cryo transfer?'
- Page 5, line 257-258, it has been reported before. (Xu et al *Structure* **2018**)

- Page 6, line 294-295, What is the successful rate of milling crystal lamellae embedded in LCP layers? Based on Sup Figure 1, the disagreement between experimental depth and iFLM estimated depth is still quite significant. In some cases, it can be as large as $\sim 15 \mu\text{m}$. A microcrystal could easily be missed or milled away.
- In Figure 1, what is the large bright contrast to the right of the selected crystal? Is it a large crystal? If so, why didn't the author choose to mill the large crystal instead as the chance of missing it would be lower. As seen in Figure 1D, a large chunk of the large crystal (if it is a crystal) was not on Cu grid bar.
- Is there a relationship between the thickness of the LCP layer and the detectability of crystals by iFLM? It seems that only crystals embedded in relatively thin LCP layer (near the edge of the spread) could be detected.
- How many crystals can the authors typically find in LCP layer using iFLM?
- In Figure 5, why only a small part of the crystal was milled into the lamella? Was it on purpose or was the target miss aligned?
- I am just curious, why didn't the authors collect MicroED data of $A_{2A}AR$ crystals using the superior Krios + Falcon 4i setup?
- All letters in space groups should be italic, screw axes should also be written properly.

Table R1 – MicroED structures of Proteinase K determined from FIB milled crystal lamellae in Gonen Lab

FIB type	Argon	Xenon	Gallium	Nitrogen	Oxygen	Best Merge	Gallium	Gallium
Citation	This work	This work	Martynowycz Nat. Methods 2022	This work			Martynowycz PNAS 2021	Clabbers J. Struct. Bio. 2022
Microscope	Krios	Krios	Krios	Krios	Krios	Krios	Talos	Krios
Detector	Falcon 4i	Falcon 4i	Falcon 4i	Falcon 4i	Falcon 4i	Falcon 4i	CETA-D	K3
Acc. Voltage	300	300	300	300	300	300	200	300
Wavelength	0.0197	0.0197	0.0197	0.0197	0.0197	0.0197	0.0197	0.0197
Resolution	19.77-1.4 (1.45-1.40)	19.74-1.45 (1.50-1.45)	43.35-1.5 (1.554-1.5)	19.75-1.8 (1.86-1.8)	20.36-1.5 (1.55-1.5)	20.63-1.39 (1.44-1.39)	High res 2.1	43.40-1.70 (1.73-1.70)
Space Group	P4₃2₁2	P4₃2₁2	P4₃2₁2	P4₃2₁2	P4₃2₁2	P4₃2₁2		P4₃2₁2
Unit Cell	67.02, 107.53	67.05, 107.02	67.08, 106.78	67.12, 106.87	67.26, 106.81	67.02, 107.53		67.01, 106.56
Total Ref.	1231493 (96739)	925857 (66300)	416133 (36794)	354396 (34224)	529482 (41429)	2726169 (150232)		598583 (11074)
Unique Ref.	47738 (4616)	43468 (4250)	39347 (3683)	23288 (2268)	38542 (3742)	49781 (4770)		27211 (1289)
Multiplicity	25.8 (20.7)	21.3 (15.5)	10.6 (9.9)	15.2 (15.0)	13.7 (11.0)	54.8 (31.3)	3.1	22 (8.6)
Completeness	97.27 (96.17)	98.61 (98.29)	98.87 (94.41)	99.68 (99.21)	96.20 (95.51)	99.44 (97.05)	82.5	99.3 (91.3)
Mean I/Sigma	7.61 (1.40)	7.73 (1.62)	5.65 (1.12)	4.18 (1.24)	4.91 (1.03)	10.78 (1.68)	3.61	4.8 (1.1)
Wilson B-factor	11.9	12.65	13.16	18.53	9.69	11.71		
R-merge	0.3235 (1.784)	0.2942 (1.475)	0.277 (1.508)	0.5183 (1.798)	0.759 (1.674)	0.3396 (1.773)		0.638 (1.574)
R-meas	0.3301 (1.829)	0.3008 (1.525)	0.291 (1.59)	0.5376 (1.862)	0.7844 (1.757)	0.3426 (1.802)	0.357	0.653 (1.675)
R-pim	0.06426 (0.3945)	0.0608 (0.3758)	0.08711 (0.4921)	0.1379 (0.4721)	0.1944 (0.5264)	0.04454 (0.3109)	0.144	0.134 (0.551)
CC1/2	0.993 (0.266)	0.995 (0.306)	0.989 (0.31)	0.964 (0.299)	0.877 (0.274)	0.997 (0.327)	0.964	0.972 (0.108)
Reflection used in Ref.	47632 (4616)	43393 (4250)	39301 (3683)	23271 (2268)	38471 (3742)	49735 (4770)		25625
Reflection for Rfree	2329 (242)	2167 (207)	2001 (210)	1142 (101)	1965 (211)	2476 (227)		1286
R-work	0.1374 (0.2720)	0.1387 (0.2869)	0.1495	0.1679 (0.2844)	0.1634 (0.2737)	0.1192 (0.2780)	0.199	0.176

R-free	0.1735 (0.3108)	0.1770 (0.3488)	0.2046	0.2121 (0.3796)	0.2138 (0.3431)	0.1634 (0.2964)	0.245	0.254
Macromolecules	2063	2052	2029	2031	2047	2031		
Ligands	10	10	37	2	10	6		
Solvent	307	294	234	237	344	322		
Residues	279	279	279	279	279	279		
RMS (bonds)	0.015	0.009	0.015	0.004	0.002	0.016	0.01	0.012
RMS (angles)	1.1	0.88	1.89	0.63	0.48	1.84	0.88	1.430
Rama. favoured	97.11	97.47	97.11	96.39	97.47	96.75		93.50
Rama. Allowed	2.89	2.53	2.89	3.61	2.53	3.25		5.78
Rama. Outliers	0	0	0	0	0	0		0.72
Rotmer Outliers	0.91	0	0.94	0	0	0		0.36
Clashscore	2.47	2.24	4.99	5.3	1.99	3.02		
Ave. B-factor	14.47	15.14	16.12	19.42	12.16	14.77		15.89
Macromolecules	12.51	13.43	14.19	18.47	10	12.77		
Ligands	27.44	25.25	46.67	20.14	24.69	20.47		
Solvent	27.21	26.76	28.02	27.49	24.63	27.29		

Table R2 – MicroED structures of A_{2A}AR determined from FIB milled crystal lamellae in Gonen Lab

FIB type	Xenon	Gallium
Citation	This work	Martynowycz et al PNAS 2021
Sample Type	LCP	Sponge phase
Setup	Talos + CETA-D	Krios + CETA-D
Acc. Voltage	200	300
Wavelength	0.0251	0.0197
Resolution	38.13 – 2.703 (2.8 – 2.703)	37.91 – 2.794 (2.894 – 2.794)
Space Group	C222 ₁	C222 ₁
Unit Cell	39.04, 177.51, 137.9	40.0, 180.5, 139.7
Total Ref.	41578 (4038)	37130 (3584)
Unique Ref.	9646 (698)	
Multiplicity	4.3 (4.2)	3.7 (3.7)
Completeness	65.69 (51.47)	77.07 (72.32)
Mean I/Sigma	3.32 (0.75)	7.46 (1.31)
Wilson B-factor	50.01	55.66
R-merge	0.3632 (1.235)	
R-meas	0.4174 (1.149)	
R-pim	0.196 (0.6685)	0.1879 (0.7499)
CC1/2	0.932 (0.233)	0.923 (0.15)
Reflection used in Ref.	8974 (698)	
Reflection for Rfree	419 (37)	
R-work	0.2561 (0.3321)	0.2482 (0.3354)
R-free	0.2971 (0.3009)	0.2881 (0.3852)
Number of non-H atoms	3117	3140
Macromolecules	3105	
Ligands	0	
Solvent	12	
Residues	390	390
RMS (bonds)	0.002	0.002
RMS (angles)	0.41	0.41
Rama. favoured	97.67	97.41
Rama. Allowed	2.33	2.59
Rama. Outliers	0	0
Rotmer Outliers	0.92	1.94
Clashscore	4.45	3.6
Ave. B-factor	43.98	43.1
Macromolecules	43.99	43.24
Ligands		40.45
Solvent	40.93	31.11

Response to reviewer comments:

We would like to thank the editor and reviewers for their careful consideration of our manuscript. We are pleased that all reviewers found our work to be of high quality and importance. Our manuscript has been revised taking all reviewers' comments into account. We believe the revised manuscript is now ready for acceptance in Nature Communications.

Below, we present a point-by-point response to all the reviewers' questions and comments. The original comments are indented in black italics and our responses follow in blue. Similar comments have been grouped where possible.

Reviewer #1

“A fundamental issue with attempting ion-beam milling of LCP embedded crystals is that this material is exceptionally difficult to mill using a gallium beam. Typically, the LCP will begin to indent, turn black, and then deform rather than being removed from the sample. Milling under these conditions is essentially impossible and has prevented milling into thicker LCP areas on the grids. Finding a method to mill away LCP without changing the phase requires a new approach to milling thick samples that does not involve a standard gallium ion beam.”

There is no reference associated with this statement nor images within this manuscript. Given the technical nature associated with developing plasmas to ameliorate this observation, even if anecdotally well documented in the field, the results outlined herein would benefit from direct comparisons.

Response: We agree but it's a statement based on experience. We have added a new **Supplementary Figure 1** to demonstrate some of our previous attempts.

All three reviewers (**#1**, **#2**, and **#3**) shared similar comments regarding alternate identification methods:

The authors detail methods using fluorescently-labeled protein in LCP. However, this may not always be applicable and could be seen as a critical limitation for such a robust approach. Notably, discussion pertaining to this potential limitation is lacking in the manuscript. What if the protein cannot be labeled prior to crystallization? Would small molecule crystal soaks suffice? Is tryptophan fluorescence insufficient due to poor quantum yield? Are there alternative methods that can be explored? What are the size limitations of the crystals that can be observed using this method?

There are other ways to detect protein in crystals, for example, UV imaging and second order nonlinear imaging of chiral crystals (SONICC), even though those may not be as specific as fluorescent labeling. Are these methods viable as possible alternatives for LCP protein crystal detection in a pFIB/SEM machine?

Page 3, Line 148-149, will tagging protein with a fluorophore prior to crystallization affect the crystallization process in some cases? Could fluorophore tags be added to microcrystals instead?

Response: We have added new data to the manuscript that shows the use of an additional fluorophore (**Supplementary Figure 18**) and a much higher resolution final A_{2A}AR structure. Our discussion now covers limitations and possibilities using other methods such as UV/SONICC and external fluorescence methods (Line 365 – 373).

Figure 3- I think it would benefit to show exemplar images from the SI in the main figure since so much discussion pertains to the quality of lamellae produced and their impacts on final data and structure quality.

Response: All good lamellae look essentially the same and remain in the SI. However, the revised manuscript now has a new **Figure 4** that shows some of the observed pathologies seen in each gas as opposed to only showing examples of the good lamellae as seen in **Figure 3**.

Reviewer #1 and **reviewer #3** shared a comment regarding the microscopes and cameras used in this work.

Not of much significance but was the A2AAR structure determined on an Arctica compared to a G3i for the proteinase K structures out of convenience (likely)? Or other reasons that are worth discussing?

I am just curious, why didn't the authors collect MicroED data of A2AAR crystals using the superior Krios + Falcon 4i setup?

Response: At the time of the original data collection, the Krios was unavailable. Now, new data have been collected and incorporated in this manuscript. This is present in the main text, **Figure 6** and **Figure 7**, and the Methods sections.

This new data demonstrate a marked improvement in attainable resolution.

Reviewer #1 and **reviewer #2** shared similar comments regarding the A₂AAR adenosine structure presented in our work.

Figure 5- This figure would greatly benefit from images of the density for different regions of A2AAR with their regions in panel D highlighted (like Figure 4A). At 2.7 Å resolution the quality of side chain density can vary widely. Alone, panel D does not provide much information besides what a GPCR looks like (which is common knowledge at this point).

The paragraph beginning on line 301 describes MicroED data collection, processing, and structural analysis of the GPCR lamella, and makes a comparison with their previous structure reported in Martynowycz et al. 2021 (PNAS). The refined density map of the A2AAR structure is not shown or uploaded as a supplement in this study, though it was described in the text. It would be helpful to the readers to see this data, particularly the refined density map, for comparison with the previous structure (of data collected after conversion to the sponge phase), and also since density maps were shown of the proteinase K data in this work (Figure 4).

Response: The revised manuscript has an improved A₂AAR structure at considerably better resolution. Density is now shown in **Figure 7**.

Reviewer #2

It would be interesting to compare SRIM simulations of different plasma beam sources on 200-300 nm-thick protein crystal lamellae with the experimental results, as well as with the previous report of SRIM simulations on tungsten metal, to guide further characterization of the ideal pFIB-SEM setup.

Response: We agree. The revised manuscript includes SRIM calculations from both tabulated values and Monte Carlo simulations (**SI Figure 8**).

Please address why the A2AAR dataset was indexed in DIALS vs XDS (line 566), since XDS worked for indexing proteinase K. XDS could not find the indexing solution?

Response: All data in the revised manuscript have now been processed using XDS. There was an error in the original input file that prevented the data from indexing properly. Once fixed, XDS worked as intended (Lines 611 - 612).

Reviewer #3

Data statistics and key structure refinement indicators reported in this work show no significant improvements over the previous reported work. The small differences in R_{work} and R_{free} values are probably due to other reasons such as the multiplicity of the data. Moreover, when studying the structure models and maps obtained from lamella prepared by each plasma source and the prior best gallium structure (Martynowycz Nat. Methods 2022) shown as figure 4 in the manuscript, I could not see huge differences between them, apart from maybe the slight resolution reduction of the map in the Nitrogen case.

Nevertheless, one of the benefits of using pFIB milling seems to be the faster milling speed, the ability to mill through thicker sample, and lower implantation depth.

Response: We rewrote to clarify the claims. The largest benefit is the ability to use higher currents without losing beam coherence. This allows us to make lamellae much faster. This is critical to milling through thick material, such as LCP. The reduced damage is an added benefit.

An additional example where the authors could clearly show the advantages of milling directly from LCP is needed to support their claim. Furthermore, milling crystals embedded in thick LCP layers is technically demanding and less available comparing to conventional cryo-FIB by Gallium ion.

Response: We have added additional data on A_{2A}AR from plasma-milled lamellae. These were prepared using an additional fluorophore and are of considerably higher resolution.

Why did the authors test the effects of different ion sources using crystals embedded in vitrified ice, if they want to study direct pFIB milling of crystals in LCP (Sup Figure 2)? Why didn't they test different ion sources using for example proteinase K crystals embedded in LCP layers? It would provide direct evidence on how each source would perform on LCP milling. It will also be more reasonable to find the best milling conditions using such sample.

Response: Prior to our experiments on vitrified crystals, it was not known that plasma milling would work at all on biological samples. The experiments on proteinase K were necessary first steps, as this work is the first to demonstrate plasma milling from frozen, biological material. We agree that future work should investigate the nuance of plasma-LCP interactions.

Page 2, Line 85-87, some early work on FIB milled lamella for MicroED were not cited: Duyvesteyn et al. PNAS 2018 (first report on FIB milled crystal lamella for MicroED), Li et al. Biophysics Reports 2018 (doi: 10.1007/s41048-018-0075-x) Zhou et al. Journal of Structural Biology 2019 (doi: 10.1016/j.jsb.2019.02.004)

Response: We have added these references.

Page 3, line 109-110, the improved signal-to-noise ratio is probably attributed to the increased multiplicity.

Response: As now stated in the text, this is certainly a factor, but not necessarily the dominant contributor. For example, the table provided by the reviewer shows the multiplicity for both nitrogen and oxygen to be higher than gallium, but with lower overall I/σ values worse refinement statistics. We have added discussion about this caveat to the manuscript and outline that this is likely an additional factor.

Page 3, Line 115, I would not call the iFLM imaging on-the-fly. It is more like switching between pFIB milling and iFLM inspection. (also on page 6 line 290-291, page 7 line 350-351). On-the-fly means imaging while milling.

Response: We agree. This phrase has been removed.

Page 4, Line 159, what is the accuracy of the overlay? I would assume for milling microcrystals, it should have sub-micrometer accuracy.

Response: The image-to-image RMSD is now given in the figure legends and SI, and we now discuss some of the limitations in the correlation process in the manuscript.

Page 5, Line 212, are there any evidences that 'ice contaminations and breakage not observed in the SEM prior to loading in the TEM are attributed to the cryo transfer?'

Response: Yes. We now directly point out a few of these cases in the manuscript. However, it was in a relatively small number of instances. See new Figure 4.

Page 5, line 257-258, it has been reported before. (Xu et al Structure 2018)

Response: We have added this reference.

Page 6, line 294-295, What is the successful rate of milling crystal lamellae embedded in LCP layers? Based on Sup Figure 1, the disagreement between experimental depth and iFLM estimated depth is still quite significant. In some cases, it can be as large as ~15 μm. A microcrystal could easily be missed or milled away.

Response: The old SI Figure 1 is not comparable to milling crystals. The beads often fly off the sample or embed deeper during milling. We did this as a sanity check to characterize that the measured distances between the iFLM, SEM, and pFIB were reasonable prior to working on embedded crystals. This also allowed

us to calculate the PSF from buried material that was necessary for later deconvolutions. In general, we take intermediate images between milling steps to reassess that we do not miss the crystal. It is relatively easy to identify over-milling, as the shape of the fluorescent outline changes as the crystal is milled. These details and nuances have been added to both the main text and in considerable detail in the methods sections.

In Figure 1, what is the large bright contrast to the right of the selected crystal? Is it a large crystal? If so, why didn't the author choose to mill the large crystal instead as the chance of missing it would be lower. As seen in Figure 1D, a large chunk of the large crystal (if it is a crystal) was not on Cu grid bar.

Response: This is a corner/edge. These reflect very brightly in any selected channel. The method of selecting a real crystal from such a feature is done by illuminating with a wavelength with no expected fluorescence and seeing if the area is still bright. This point and method have been added to the methods and main text to be clearer. The smaller crystal to the right of the selected crystal does overlap the Cu grid bar, which is why it was not milled.

Is there a relationship between the thickness of the LCP layer and the detectability of crystals by iFLM? It seems that only crystals embedded in relatively thin LCP layer (near the edge of the spread) could be detected.

Response: Crystals can be detected even in very thick LCP layers. This is accomplished by acquiring a Z-stack through the LCP layer. However, the signal becomes less sharp with increasing depth. A large amount of the haze/noise comes from out-of-focus scattering. Z-stack deconvolution allows for more accurately targeting even deep crystals but is computationally expensive. This is now discussed in greater detail in the manuscript.

How many crystals can the authors typically find in LCP layer using iFLM?

Response: In our experience the limiting factor is seldom the number of crystals, but the large amount of time the operator must spend obtaining, correlating, and then Z-projecting the images between the SEM, FIB, and iFLM to determine if a crystal can be suitably milled without overlapping with a grid bar or the milling beam being blocked by other piles of LCP at a grazing incidence. Since the transfer of crystals is currently from a glass sandwich plate to an EM grid, the number depends on the number in the initial well and the effectiveness of the operator. We have added this caveat to our discussion.

In Figure 5, why only a small part of the crystal was milled into the lamella? Was it on purpose or was the target miss aligned?

Response: This area was specifically selected to avoid the nearby grid bar mentioned earlier. The lamellae added in our revision are better centered, since they were not as close to grid bars.

All letters in space groups should be italic, screw axes should also be written properly.

Response: We have corrected these errors.

Reviewers' Comments:

Reviewer #1:

Remarks to the Author:

I have no further request for this manuscript. However, the figure 7 PDF file is blank. If the authors provided panels in figure 7 in line with the figure legend then I am satisfied.

Reviewer #3:

Remarks to the Author:

I would like to thank the authors for addressing all reviewers' comments, performing additional experiments and revising the text, which clearly improved the quality of the manuscript. I appreciate the authors' efforts in improving crystal identification by milling holes as reference points. Furthermore, with additional data and results of the A2AAR crystals, I am convinced that the new methods developed by the authors could improve MicroED data and structure quality. Therefore, I strongly support the manuscript to be published in Nature Communications.

A few minor points:

- Figure 4 seems to be missing in the merged file.
- Line 108-109, maybe the authors could already refer to Sup Figure 8.
- Line 606 – 608, the sentence '... using the same camera and rotation rate as the proteinase K crystals using the Falcon 4 direct electron detector operating in electron counting mode.' The sentence is a bit confusing. I assumed that the authors meant 'two additional data of A2AAR crystals were collected using the Falcon 4'.
- In the data processing part (starting from line 610), it would be nice if the author can clarify that the final A2AAR data was produced by merging 1 dataset collected on Talos + Ceta-D and 2 datasets collected on Krios + Falcon 4. Being able to merge data collected on different platforms suggest the method is robust.